# DiffInk: Glyph- and Style-Aware Latent Diffusion Transformer for Text to Online Handwriting Generation

**Wei Pan, Huiguo He**[*]**, Hiuyi Cheng, Yilin Shi, Lianwen Jin**[*]
South China University of Technology
eewpan@mail.scut.edu.cn, {hehuiguo, eelwjin}@scut.edu.cn

## Abstract

Deep generative models have advanced text-to-online handwriting generation (TOHG), which aims to synthesize realistic pen trajectories conditioned on textual input and style references. However, most existing methods still primarily focus on character- or word-level generation, resulting in inefficiency and a lack of holistic structural modeling when applied to full text lines. To address these issues, we propose **DiffInk**, the **first** latent diffusion Transformer framework for full-line handwriting generation. We first introduce **InkVAE**, a novel sequential variational autoencoder enhanced with two complementary latent-space regularization losses: (1) an *OCR-based loss* enforcing glyph-level accuracy, and (2) a *style-classification loss* preserving writing style. This dual regularization yields a semantically structured latent space where character content and writer styles are effectively disentangled. We then introduce **InkDiT**, a novel latent diffusion Transformer that integrates target text and reference styles to generate coherent pen trajectories. Experimental results demonstrate that DiffInk outperforms existing state-of-the-art (SOTA) methods in both glyph accuracy and style fidelity, while significantly improving generation efficiency.

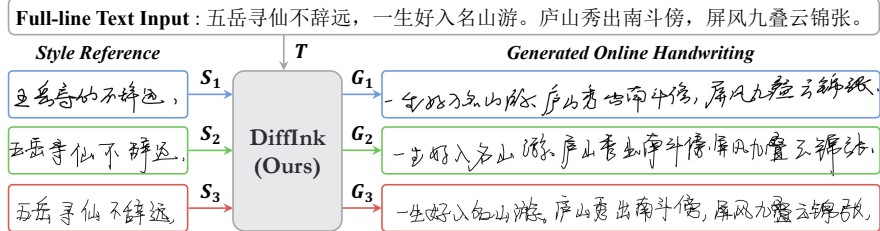

Figure 1: **Overview of DiffInk**. By directly modeling entire text lines rather than individual characters, the model efficiently synthesizes online handwritten text lines ($G_i$) conditioned on textual input ($T$) and style references ($S_i$), achieving accurate content reproduction and consistent style in both character form and layout structure. Different colors represent distinct handwriting styles.

## 1 Introduction

As shown in Figure 1, TOHG refers to the task of synthesizing realistic pen trajectories conditioned on textual content and style references. It has a wide range of applications, including personalized digital ink rendering, handwriting simulation, and data augmentation for optical character recognition (OCR). However, TOHG presents unique challenges: beyond character-level generation, it requires coherent inter-character modeling and line-level layout; Compared to image-based methods, it needs to handle temporal dynamics while maintaining glyph fidelity and stylistic consistency.

In recent years, TOHG has attracted growing research interest. Early methods (Graves, 2013; Li et al., 2024) adopt Auto-Regressive (AR) modeling to generate trajectory points sequentially. Subsequent approaches attempt to control style through content-style disentanglement, utilizing either 1) coarse-grained global style representations such as writer IDs (Luo et al., 2022) or 2) fine-grained local style representations such as character components (Liu et al., 2024; Dai et al., 2023). Notably,

---

[*]Corresponding authors. Code available at https://github.com/awei669/DiffInk.

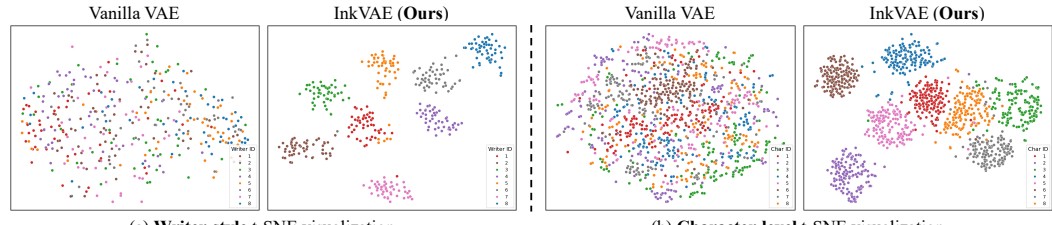

Figure 2: **Latent-space visualization of Vanilla VAE vs. InkVAE (ours).** While both models achieve good reconstruction, InkVAE learns a more structured latent space (visualized with t-SNE (Maaten & Hinton, 2008)): (a) Text-line features from 8 writers—InkVAE exhibits clearer writer-specific clusters. (b) Character-level features from 8 common characters—InkVAE yields tighter intra-class groupings and more distinct inter-class separation.

OLHWG (Ren et al., 2025) adopts a novel diffusion-based framework to generate isolated characters and arranges them using external layout prediction modules, achieving SOTA performance. *However, these methods all fundamentally **operate at the character or word level**, lacking the ability to capture long-range dependencies and overall text-line structure.*

To tackle these limitations, we propose **DiffInk**, a novel glyph- and style-aware conditional latent diffusion Transformer for TOHG. DiffInk begins by training a sequential variational autoencoder (VAE) to learn compact latent representations of full text lines, thereby avoiding character-by-character modeling and capturing long-range structural dependencies. However, although a vanilla VAE can accurately reconstruct handwriting sequences when trained with only reconstruction loss, adding auxiliary perceptual losses directly on the generated outputs yields limited benefit, as the representations it learns lack semantic structure. Specifically, the VAE fails to group features by writer identity or character label in its internal representation space. As illustrated in the left panels of Figure 2 (a) and (b), features from different writers or characters overlap in an unstructured manner. As a result, small perturbations introduced by the diffusion model in this space can easily push the generated output toward an incorrect writing style or an unintended character.

To resolve this, we introduce two lightweight regularization losses to regulate semantic structure in the latent space: (1) an OCR-based loss to ensure glyph-level accuracy and (2) a style-classification loss to enforce writer-level consistency. These losses are implemented by two trainable classifiers directly applied in the latent space and jointly optimized with the VAE reconstruction loss. We refer to the resulting enhanced VAE as **InkVAE**, which learns a well-structured latent space where glyph and style features are both effectively disentangled, enabling precise control over content and style generation. As evidenced in the right panels of Figure 2 (a) and (b), this regularization substantially restructures the latent space, yielding distinct clustering by both character and writing style. This regularized latent space offers a robust foundation for conditional generation.

Building on this regularized latent space, we then develop a novel **InkDiT** that synthesizes full-line handwriting conditioned on both target text sequences and short reference trajectories. The InkDiT denoises latent representations concatenated with textual content and style features, predicting clean outputs through iterative refinement. Specifically, textual content features are obtained by sequentially mapping characters to embeddings from a learnable codebook, followed by a lightweight content encoder. Style features are extracted from the reference trajectories using our InkVAE encoder. These features encompass both local attributes (e.g., character shapes and strokes) and global attributes (e.g., spacing and alignment). Subsequently, InkDiT effectively leverages these conditions to generate content-accurate and style-consistent results.

Our main contributions can be summarized as follows: (1) We propose **DiffInk**, the **first** latent diffusion framework for end-to-end full-line handwriting generation, capable of generating glyph-accurate and style-consistent pen trajectories. (2) We introduce **InkVAE**, a sequential VAE enhanced with two task-relevant lightweight regularization losses (*OCR-based loss* and *style-classification loss*) to disentangle content and style in the latent space, enabling structured representation learning. (3) We propose a novel **InkDiT** that jointly conditions on target text and reference style, generating realistic handwriting trajectories via an iterative denoising process. (4) Experiments on the CASIA-OLHWDB 2.0–2.2 benchmark demonstrate that DiffInk significantly outperforms current SOTA methods in full-line generation quality, while also offering improved efficiency.

## 2 RELATED WORK

### 2.1 ONLINE HANDWRITING GENERATION

Early methods for online handwriting generation were predominantly RNN-based. Approaches such as Graves' seminal work (Graves, 2013) and DeepWriting (Aksan et al., 2018) adopted LSTMs with mixture density networks to model temporal dynamics, but lacked explicit style control. SketchRNN (Ha & Eck, 2017) used an RNN-based VAE to generate ink data unconditionally. More recently, TrInk (Jin et al., 2025) introduced a Transformer-based architecture, which leverages global attention to generate more realistic and coherent handwriting trajectories.

In contrast to Latin-based scripts, languages like Chinese possess a vast set of characters with highly complex structures and stroke compositions, which has led to the proposal of specialized approaches to capture these intricacies. FontRNN (Tang et al., 2019) incorporated monotonic attention for stroke-level synthesis, while CHWmaster (Li et al., 2024) utilized a sliding-window RNN for few-shot personalized generation. SDT (Dai et al., 2023) employed dual-branch encoders with contrastive learning to disentangle writer style from character content, and WLU (Tang & Lian, 2021) applied meta-learning to support rapid personalization. Building on the diffusion paradigm, DiffWriter (Ren et al., 2023) formulated character-level synthesis as a conditional denoising process, and OLHWG (Ren et al., 2025) further extended this framework by decoupling glyph synthesis and layout planning, thereby enabling layout-aware generation of full handwritten text lines. *However, these approaches essentially remain focused on **modeling individual glyphs***, with limited efforts toward directly modeling entire text lines. While OLHWG assumes that character generation is independent of layout position and attempts to decouple a text line into isolated characters and layout modules, such assumptions fail to capture the complexity of real handwriting. Appendix D.1 further discusses our distinctions from existing character-level and layout-decoupled approaches.

### 2.2 STYLE TRANSFER FOR HANDWRITING GENERATION

Controlling both content and style remains a central challenge in handwriting generation. For content representation, most methods employ content encoders to extract structural information from template inputs (Pippi et al., 2023). Some approaches (Pan et al., 2023; Kang et al., 2021) further design fine-grained encoders to capture detailed local patterns. Alternatively, several methods (Ren et al., 2023; 2025) adopt learnable character embeddings to provide high-level semantic information. For style representation, early approaches (Kang et al., 2020; Gui et al., 2023) employed learnable style embeddings to represent the writing style of specific authors. Later methods (Dai et al., 2023; Liu et al., 2024) introduced style encoders to extract style features from a few reference samples, using mechanisms such as cross-attention or adaptive normalization (Huang & Belongie, 2017) to guide the generation process and maintain stylistic consistency. (Kotani et al., 2020) obtains style representations via matrix factorization. For text-line handwriting generation, using a continuous text-line reference (Li et al., 2024; Pippi et al., 2025) has become a popular approach to encode local writing style and overall layout.

### 2.3 DIFFUSION GENERATIVE MODELS

Diffusion models (Ho et al., 2020) have emerged as powerful tools in generative modeling, demonstrating impressive performance across images (Rombach et al., 2022), speech (Lee et al., 2025), and vision-language tasks (Luo et al., 2023). To reduce computational costs and enhance semantic control, many recent works (Zhang et al., 2023; Saharia et al., 2022; Ramesh et al., 2022; Labs, 2024) adopt latent diffusion models, where the denoising process is conducted in a learned latent space. Recently, Diffusion Transformers (DiT) (Peebles & Xie, 2023) have replaced traditional U-Net (Ronneberger et al., 2015) backbones with Transformer-based (Vaswani et al., 2017) architectures, enabling stronger global conditioning in image synthesis tasks. Variational Autoencoders (VAEs) (van den Oord et al., 2017; Baevski et al., 2020) are commonly used to construct such spaces, offering compact and structured representations.

Overall, while considerable progress has been made in handwriting generation and latent diffusion modeling, generating complete online handwritten text lines remains a challenging and underexplored task due to long-range dependencies and complex layouts. To this end, we propose DiffInk, a structure-aware generative framework based on a latent diffusion Transformer, providing a promising direction for tackling text-to-online handwriting generation.

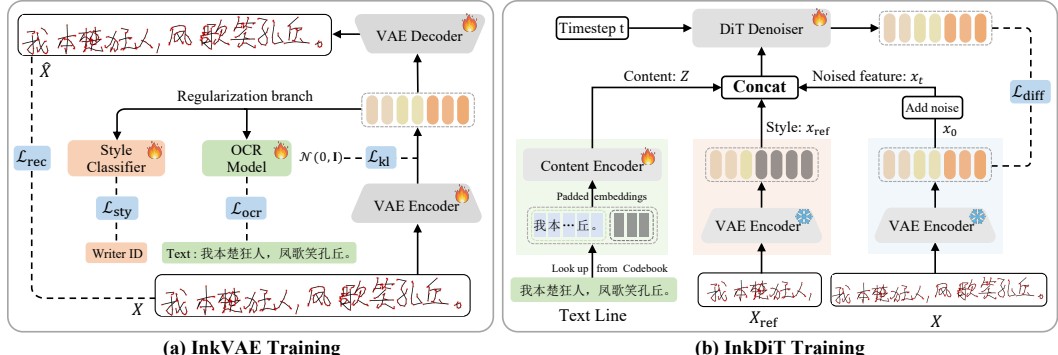

Figure 3: **Overview of the DiffInk Framework.** (a) InkVAE encodes online handwriting sequences into compact latent representations. During training, regularization losses $\mathcal{L}_{\text{ocr}}$ and $\mathcal{L}_{\text{sty}}$ are applied to the latent space to encourage disentangled glyph and style. (b) InkDiT leverages this latent space to synthesize handwriting by denoising noisy inputs $x_t$ into clean representations $x_0$. The process is conditioned on content features $Z$ obtained from text embeddings and style features $x_{\text{ref}}$ derived from a reference trajectory. InkDiT is trained with a diffusion loss $\mathcal{L}_{\text{diff}}$.

## 3 METHOD

### 3.1 PRELIMINARY

**Online Handwriting Data**   An online handwriting text-line data is represented as a pair $(T, X)$, where $T = \{t_1, t_2, \ldots, t_m\}$ denotes a sequence of $m$ characters, and $X = \{\text{seq}_1, \text{seq}_2, \ldots, \text{seq}_m\}$ is the corresponding sequence of pen trajectories. Each $seq_i \in \mathbb{R}^{n_i \times 5}$ encodes the trajectory of character $t_i$ with $n_i$ points. The total number of points for the entire line is $\sum_{i=1}^{m} n_i$. Each point is represented as a 5-dimensional vector $(x, y, pen)$, where $(x, y)$ are spatial coordinates and $pen$ is a one-hot vector indicating the pen state: *Pen Down*, *Pen Up*, or *End of Character*.

**Task Overview**   Compared with single-character handwriting generation, TOHG is considerably more complex and challenging. First, text-line generation often involves much longer sequences (typically over 25 characters), which substantially increases modeling difficulty. Moreover, in addition to capturing handwriting style, TOHG also involves modeling inter-character dependencies and structural relationships across the line to produce fluent and natural handwriting. To make the problem precise, we therefore formalize TOHG as follows: the input consists of a reference trajectory $X_{\text{ref}}$ and a concatenated text combining $T_{\text{ref}}$ and $T_{\text{gen}}$, and the goal is to generate a trajectory $X_{\text{gen}}$ that renders $T_{\text{gen}}$ while preserving the handwriting style in $X_{\text{ref}}$.

### 3.2 OVERVIEW OF THE DIFFINK FRAMEWORK

As illustrated in Figure 3, DiffInk consists of two key components: InkVAE, a pre-trained sequential variational autoencoder, and InkDiT, a conditional latent diffusion Transformer. InkVAE compresses online trajectories $X \in \mathbb{R}^{N \times 5}$ into latent representations $x \in \mathbb{R}^{l \times d}$, capturing both glyph-level structures and writer-specific style features. InkDiT operates within this structured latent space. It receives a noisy latent input $x_t$ along with two conditioning signals: (1) a style prompt $x_{\text{ref}} \in \mathbb{R}^{l \times d}$, obtained by encoding the reference trajectory $X_{\text{ref}}$ with the InkVAE encoder; and (2) a content condition $Z \in \mathbb{R}^{l \times d_{\text{text}}}$, encoded by the content encoder from text embeddings. Guided by these conditions, InkDiT denoises $x_t$ to reconstruct the clean latent representation $x_0$. During inference, the process starts from pure Gaussian noise and, under the same content and style conditions as in training, applies DDIM sampling to obtain a clean latent representation. This refined latent is then decoded by the InkVAE decoder to generate the final handwriting sequence.

### 3.3 GLYPH- AND STYLE-AWARE INKVAE

As shown in Figure 3 (a), InkVAE employs an encoder–decoder architecture, primarily designed for efficient modeling of handwriting sequences through a compact latent space. Beyond this, InkVAE incorporates a novel task-relevant regularization strategy to construct a well-structured latent space,

which promotes the disentanglement of content and style, leading to more robust representations across both diverse handwriting styles and character classes. Such structured representations, as evidenced by recent studies (Tang et al., 2025; Yao et al., 2025; Guo et al., 2025), have been shown to improve the convergence efficiency and overall performance of downstream diffusion models.

**Trajectory Encoder**    To extract complex handwriting features, we employ a 1D convolutional encoder to transform the input handwriting sequence $X \in \mathbb{R}^{N \times 5}$ into a compact latent representation $x \in \mathbb{R}^{l \times d}$. This representation is directly provided as input to three branches: (1) a trajectory decoder for handwriting reconstruction, (2) a Transformer-based OCR module for text-line handwriting recognition, and (3) a style classifier for writer identification.

**Trajectory Decoder**    Following (Tang et al., 2019), the decoder first applies a 1D convolutional stack that mirrors the trajectory encoder, up-sampling the latent feature $x$ into a feature map $O_t \in \mathbb{R}^{N \times (6p+3)}$. The feature vector $O_t$ is divided into two branches: (1) $6p$ logits that parameterize a $p$-component Gaussian Mixture Model (GMM) (Reynolds, 2015) for predicting the $(x, y)$ coordinates, and (2) 3 logits for classifying the pen state $pen$. Accordingly, the reconstruction loss is defined as $\mathcal{L}_{\text{rec}} = \mathcal{L}_{\text{pen}} + \mathcal{L}_{\text{gmm}}$, where $\mathcal{L}_{\text{pen}}$ is a focal loss used for the three-way pen-state classification, and $\mathcal{L}_{\text{gmm}}$ is the negative log-likelihood of the ground-truth points under the predicted mixture. To enable the model to learn when to stop writing, the GMM loss is computed only over valid trajectory steps, while the pen-state loss is applied across the entire sequence, including padded regions.

**Glyph- and Style-Aware Regularization**    To encourage the latent space to preserve character-level structural information, we introduce a lightweight OCR module as a structural regularizer. Operating at the feature-token level, this module promotes consistency in the representations of the same character class across different writing styles. Specifically, the OCR module decodes latent representations into character sequences using a Transformer-based recognition head, and is trained with a CTC-based (Graves et al., 2006) loss, denoted as $\mathcal{L}_{\text{ocr}}$. This supervision imposes structural constraints on the encoder, promoting the learning of character-invariant features and enhancing both the structural awareness and semantic disentanglement of the latent space.

On the other hand, to enhance the discriminability of writing styles in the latent space, we introduce a lightweight style encoder as a global discriminative supervisor. This module processes the entire latent representation to extract a holistic style representation. It summarizes the latent features using an LSTM network (Staudemeyer & Morris, 2019) with attention pooling and is optimized by a writer classification loss ($\mathcal{L}_{\text{sty}}$), which explicitly encourages style-aware separation in the latent space.

**VAE Loss Function**    InkVAE is trained end-to-end with the objective function $\mathcal{L}_{\text{VAE}}$ defined in Equation 1. This training strategy enables the encoder to learn latent representations that are structurally coherent, semantically aligned, and stylistically informative, providing a solid foundation for diffusion-based handwriting synthesis. Details of InkVAE are presented in the Appendix B.1.

$$\mathcal{L}_{\text{VAE}} = \sum \lambda_\ell \cdot \mathcal{L}_\ell \quad \text{where } \ell \in \{\text{rec, kl, ocr, sty}\} \tag{1}$$

### 3.4 Handwriting Generation with InkDiT

As shown in Figure 3 (b), InkDiT employs a Transformer-based diffusion architecture for conditional online handwriting generation in the latent space. It builds on InkVAE's compact and structured representation and progressively refines noisy latent variables into coherent trajectories through denoising. By conditioning on both text-line $T_{\text{ref}} + T_{\text{gen}}$ and a reference trajectory $X_{\text{ref}}$, InkDiT achieves controllable generation with improved fidelity and continuity.

**Text-Line Content**    For content representation, each character in the input text $T_{\text{ref}} + T_{\text{gen}}$ is embedded via a learnable codebook $\in \mathbb{R}^{K \times d_{\text{text}}}$, where $K$ is the vocabulary size and $d_{\text{text}}$ is the embedding dimension. By embedding characters directly, the approach learns high-level semantic representations of characters, while avoiding the domain gap caused by the complex mapping from fixed templates to long handwritten text-line sequences (Wang et al., 2023; 2026). Text-line generation is further challenged by uncertain character lengths at inference, which prevent direct embedding–trajectory alignment as in character-level methods (Ren et al., 2023; 2025). We address this through a simple yet efficient design where character embeddings are sequentially concatenated,

padded with shared learnable embeddings to match $x_0$, and processed by a lightweight content encoder. Based on ConvNeXt-V2 (Woo et al., 2023), this encoder leverages large-kernel depthwise convolutions to capture long-range dependencies and yields a representation $Z \in \mathbb{R}^{l \times d_{\text{text}}}$ serving as semantically aligned content conditions for generation guidance.

**Text-Line Handwriting Style** For style representation, character-level approaches (Dai et al., 2023; Tang & Lian, 2021; Ren et al., 2025) typically rely on randomly sampled isolated characters from the same writer. In contrast, text-line generation requires capturing additional layout information absent in such isolated samples. To holistically model both writing style and text-line layout, we follow (Li et al., 2024; Pippi et al., 2025) and adopt a continuous handwritten trajectory $X_{\text{ref}}$ as the style reference. This trajectory is encoded into a latent representation by the trajectory encoder of our proposed style-aware InkVAE. Leveraging the VAE encoder's strength as a versatile feature extractor (Leng et al., 2025; Cheng & Yuan, 2025), we obtain the style feature $x_{\text{ref}}$. By modeling style directly within the unified latent space of the VAE (Tan et al., 2025; Chen et al., 2025), we avoid introducing a separate style encoder and ensure consistent representation learning.

**DiT Denoiser** The diffusion network follows the DiT (Peebles & Xie, 2023) architecture, leveraging its contextual modeling capacity to recover the clean latent from noisy inputs under the guidance of content features $Z$ and style features $x_{\text{ref}}$. The three inputs are concatenated along the channel dimension and linearly projected to align with the InkVAE feature space.

$$x_t = \sqrt{\bar{\alpha}_t} \cdot x_0 + \sqrt{1 - \bar{\alpha}_t} \cdot \epsilon, \quad \epsilon \sim \mathcal{N}(0, \mathbf{I}) \tag{2}$$

$$\hat{x}_0 = \text{InkDiT}_\theta([x_t, x_{\text{ref}}, Z], \ t); \quad \mathcal{L}_{\text{diff}} = \mathbb{E}_{x,t}[\text{mask} \odot |\hat{x}_\theta - x_0|^2] \tag{3}$$

DiT denoiser is trained to recover the clean feature $x_0$ at masked positions by minimizing the masked mean squared error (MSE), as defined in Equation 3, where the masked regions correspond to the reference style trajectory features. Benefiting from the semantically structured latent space provided by InkVAE, the DiT model can more effectively utilize conditional inputs to recover features from noise, resulting in better alignment with the underlying data distribution. Following the DDIM (Song et al., 2020) sampling strategy, we start from Gaussian noise and iteratively generate the final latent estimate $\hat{x}_0$, conditioned on both content and style features. Finally, this latent representation is decoded by the InkVAE decoder to reconstruct the handwriting trajectory. Details of InkDiT, including its architectural design as well as training and inference procedures, are presented in Appendix B.2.

# 4 EXPERIMENTS

## 4.1 DATASET & EVALUATION METRICS

**Dataset** We conduct experiments to validate DiffInk on Chinese handwriting datasets. *For Chinese TOHG*, we use CASIA-OLHWDB 2.0–2.2 (Liu et al., 2011), sampling 500 writers based on longer average text lines. Among them, 400 writers are used for training and 100 for testing. We then augment the training set with synthesized samples, yielding 67,000 text lines covering 2,648 characters. The test set comprises 4,780 text lines. Details are presented in the Appendix A.

**Evaluation Metrics** To comprehensively evaluate the proposed DiffInk, following prior works (Tang & Lian, 2021; Dai et al., 2023; Ren et al., 2025), we employ several widely used evaluation metrics, including: (1) Content fidelity, assessed with a pretrained text-line OCR model by reporting Accurate Rate (AR) and Correct Rate (CR) following the (Yin et al., 2013) OLHCTR benchmark; (2) Style consistency, measured with a writer classifier via writer classification accuracy (Style); (3) Trajectory similarity, evaluated using normalized Dynamic Time Warping (DTW) distance (Berndt & Clifford, 1994; Chen et al., 2022); (4) Generation efficiency, reported as the number of characters generated per second (Avg. char/s). Details are presented in Appendix B.4.

## 4.2 IMPLEMENTATION DETAILS

For Chinese TOHG, InkVAE is trained for 100 epochs with a batch size of 128 and a learning rate of $5 \times 10^{-4}$. To balance different objectives, we apply the following loss weights: $\lambda_{\text{gmm}} = 1.0$,

Table 1: **Quantitative comparison with SOTA handwriting generation methods.** Baseline methods generate isolated characters and compose text lines through a shared layout prediction module from OLHWG. In contrast, our method directly generates complete text lines and outperforms these single-character approaches across all evaluation metrics.

| Method | Venue | Output | AR% ↑ | CR% ↑ | Style ↑ | DTW ↓ | Avg. char/s ↑ |
|---|---|---|---|---|---|---|---|
| Drawing (Zhang et al., 2017) | TPAMI'17 | Char-level | 76.35 | 76.87 | 25.75 | 1.582 | 4.16 |
| DeepImitator (Zhao et al., 2020) | PR'20 | Char-level | 78.21 | 78.98 | 27.62 | 1.561 | 3.96 |
| WLU (Tang & Lian, 2021) | CGF'21 | Char-level | 79.85 | 80.23 | 35.71 | 1.540 | 25.19 |
| SDT (Dai et al., 2023) | CVPR'23 | Char-level | 82.53 | 83.00 | 50.51 | 1.270 | 3.35 |
| OLHWG (Ren et al., 2025) | ICLR'25 | Char+layout | 91.48 | 91.71 | 44.74 | 1.326 | 0.07 |
| **DiffInk** | **This work** | **Line-level** | **94.38** | **94.58** | **77.38** | **1.049** | **58.47** |

$\lambda_{pen} = 2.0$, $\lambda_{ocr} = 1.0$, $\lambda_{sty} = 0.5$, and $\lambda_{kl} = 1 \times 10^{-6}$. The DiT model is trained for 200k steps with a batch size of 256 and a learning rate of $7.5 \times 10^{-5}$. Details are presented in the Appendix B.

## 4.3 MAIN RESULTS

**Comparing with Representative Methods** As there is a lack of established baselines for end-to-end text-line generation, we adopt several SOTA character generation methods and extend them with a layout modeling module to enable line-level comparison. These baselines include Drawing (Zhang et al., 2017), DeepImitator (Zhao et al., 2020), WLU (Tang & Lian, 2021), and SDT (Dai et al., 2023), which generate high-fidelity handwritten characters under an autoregressive paradigm, as well as OLHWG (Ren et al., 2025), which synthesizes isolated characters using a diffusion-based approach. *To support text-line level comparison*, we adopt the layout prediction module from OLHWG—currently the SOTA—to compose full lines from the character-level outputs of all five methods. In the OLHWG method, the layout prediction module infers the subsequent layout based on the layouts of a fixed number of preceding characters (originally set to 10) from the ground-truth text line. To accommodate text lines of varying lengths, we instead use the layouts of the first 30% of characters from each text line as input for layout prediction (max 10). To ensure fairness, all methods are retrained on the same dataset with consistent preprocessing and default configurations.

**Quantitative Evaluation** As shown in Table 1, DiffInk (ours) achieves superior performance across all quantitative metrics. For content fidelity, the OCR-based metrics AR and CR both exceed 94%, outperforming the latest SOTA method OLHWG by an average margin of 3 percentage points, and substantially surpassing earlier approaches such as Drawing, DeepImitator, WLU and SDT—with a margin of 14.44%. For style consistency, DiffInk achieves a style score of 77.38 at the text-line level—the highest among all methods—surpassing the contrastive-learning–based OLHWG by 30 percentage points, the dual-branch style-loss–based SDT by 27 points, and baselines with simple style modeling by an average of 50 points. Moreover, style evaluation further reveals that layout stitching introduces anomalies that can be readily detected by the style classifier. In terms of structural fidelity, DiffInk attains substantially lower normalized DTW distances, indicating more consistent alignment with target trajectories. In contrast, autoregressive methods such as WLU and SDT suffer from cumulative error propagation, which severely degrades text-line level performance. Finally, DiffInk demonstrates strong generation efficiency, producing 58.82 characters per second—17× faster than SDT, over 800× faster than OLHWG, and more than 2× faster than WLU—thanks to the efficient text-line modeling of InkVAE.

**Qualitative Evaluation** As shown in Figure 4, DiffInk (ours) produces more continuous and coherent text lines with smooth character transitions. In contrast, character-wise methods such as SDT and OLHWG often exhibit unnatural stitching artifacts at character boundaries, while WLU tends to generate characters that stick together, likely due to its strong reliance on the pretrained content–style encoder. These problems arise partly because character-level methods can only assemble text lines by scaling and positioning individual characters according to their predicted bounding boxes. During inference, obtaining these bounding boxes requires an additional prediction module, which introduces its own errors and further affects the reliability of the final text-line layout. Beyond these practical limitations, although their single-character generation quality is generally good, the decoupled modeling of layout and characters neglects structural dependencies between adjacent characters, making it difficult to achieve consistent and coherent text-line layouts when character shapes differ substantially. Additional visualization results are provided in Appendix C.

| Method | Bottom-Left to Top-Right Layout | Horizontal Layout | Top-Left to Bottom-Right Layout |
|---|---|---|---|
| Text | 根据牛顿在一七零四年的一封书信上的记载 | 根据以色列《国家报》报道，牛顿在十七世纪中是最知名 | 到蓝翅蝴蝶草中，这种转基因蓝翅蝴蝶草吸磷的能力提高 |
| Drawing | | | |
| Deep Imitator | | | |
| WLU | | | |
| SDT | | | |
| OLHWG | | | |
| **DiffInk (Ours)** | | | |
| Target | | | |
| Text | "面纱"。它将于2013年接替退休的哈勃望远镜，帮助科学家观 | 据日本《东京新闻》网站22日报道说，日本研究人员从十字花科植 | 成。遥见仙人彩云间，手把芙蓉朝玉京。先期汗漫九垓上，愿接卢敖 |
| Drawing | | | |
| Deep Imitator | | | |
| WLU | | | |
| SDT | | | |
| OLHWG | | | |
| **DiffInk (Ours)** | | | |
| Target | | | |

Figure 4: **Comparison with SOTA methods** under unseen writing styles across diverse layouts. **All baseline methods generate isolated characters and compose lines via a shared layout module.** Blue boxes denote the same style reference, while red boxes highlight errors or unnatural character connections. These methods suffer from stitching artifacts, especially when adjacent characters differ structurally. DiffInk generates more coherent and naturally connected text lines.

Table 2: **Ablation study of InkVAE and its impact on InkDiT generation.** "✓" means loss enabled. While all VAEs achieve near-perfect reconstruction performance with various losses, InkDiT achieves optimal performance when trained on InkVAE latents (last row). This confirms that **our lightweight regularization losses structure the latent space for robust diffusion-based generation, with negligible impact on VAE reconstruction performance**.

| VAE Losses | | | VAE Reconstruction Performance | | | | InkDiT Generation Performance | | | |
|---|---|---|---|---|---|---|---|---|---|---|
| $\mathcal{L}_{rec+kl}$ | $\mathcal{L}_{ocr}$ | $\mathcal{L}_{sty}$ | AR% ↑ | CR% ↑ | Style ↑ | DTW ↓ | AR% ↑ | CR% ↑ | Style ↑ | DTW ↓ |
| ✓ | ✗ | ✗ | 97.59 | 97.61 | 99.97 | 0.014 | 74.77 | 75.20 | 60.68 | 1.062 |
| ✓ | ✓ | ✗ | 97.60 | 97.63 | 99.97 | 0.015 | 82.09 | 82.41 | 66.07 | 1.052 |
| ✓ | ✗ | ✓ | 97.59 | 97.62 | 99.98 | 0.016 | 79.64 | 79.95 | 68.99 | 1.059 |
| ✓ | ✓ | ✓ | 97.65 | 97.67 | 99.97 | 0.016 | **94.38** | **94.58** | **77.38** | **1.049** |

## 4.4 ABLATION STUDY OF INKVAE AND INKDIT

**Effect of InkVAE and its impact on InkDiT** As shown in Table 2, we perform an ablation study to evaluate the effectiveness of InkVAE. The left columns list the loss configurations of different VAE variants, the middle columns report reconstruction performance, and the right columns present generation results of InkDiT trained on the corresponding feature spaces. Since online handwriting data is relatively clean and less affected by background noise than image-pixel modeling, VAEs trained solely with the reconstruction loss $\mathcal{L}_{rec}$ already achieve high-fidelity reconstructions, as reflected by the low DTW distances in the middle columns. However, the diffusion results reveal a key limitation: strong reconstruction fidelity does not guarantee a well-structured or semantically meaningful latent space. Compared with the Vanilla VAE, adding OCR regularization improves recognition accuracy by about 4 points, while style regularization yields an 8-point gain in the writer-classification score. When combined—forming our proposed InkVAE—the latent space becomes markedly more suitable for diffusion modeling, as evidenced by the best overall results across recognition accuracy, style classification score, and DTW distance.

Figure 2 further visualizes the latent space. Unlike the Vanilla VAE, which fails to form clear clusters by writing style or character class, InkVAE—with OCR and style regularization—produces well-separated clusters, yielding more discriminative and semantically meaningful representations. This shows that *although the Vanilla VAE reconstructs sequences well, its latent features lack sufficient semantic information*, so even small perturbations may lead to content errors or stylistic drift during

| VAE Model | DiT generation results (trained on corresponding VAE) |
|---|---|
| Baseline | |
| + OCR loss | |
| + Style loss | |
| + OCR + Style loss (ours) | |
| Target | |

Figure 5: **InkDiT Generation with VAE Variants**. Blue boxes highlight content errors; red boxes indicate style inconsistencies. InkDiT trained on the latent space from our InkVAE yields more accurate and consistent results.

Table 3: **Ablation study of InkDiT**. The content encoder with ConvNextV2 significantly improves content accuracy and style consistency with minimal overhead, while introducing long-skip residual connections leads to performance degradation and additional computational cost.

| Model | AR% ↑ | CR% ↑ | Style ↑ | DTW ↓ | Avg. char/s ↑ |
|---|---|---|---|---|---|
| w/ Longskip | 71.73 | 71.91 | 55.42 | 1.074 | 55.55 |
| w/o ConvNextV2 | 81.51 | 81.71 | 65.49 | 1.052 | **71.94** |
| **DiffInk (Ours)** | **94.38** | **94.58** | **77.38** | 1.049 | 58.47 |

generation. Building on this, Figure 5 presents qualitative results of InkDiT trained on different VAE variants. While the Vanilla VAE often produces inaccurate content and inconsistent styles, InkVAE achieves both high content fidelity and stylistic consistency, demonstrating the effectiveness of disentangled latent spaces for controllable handwriting generation.

**Effect of InkDiT** As shown in Table 3, we perform an ablation study on InkDiT. Introducing long skip connections results in degraded performance, whereas removing the content encoder causes a marked drop in content quality and further undermines style consistency. Interestingly, even with simple padding of text embeddings and without explicit sequence-length annotations, the content encoder effectively captures long-range

| Text | 手稿展 "的希伯来大学 | 秦心之叠道初成 |
|---|---|---|
| w/o ConvNext | | |
| DiffInk(Ours) | | |
| Target | | |

Figure 6: **Ablation study on the content encoder**. Without the content encoder, content errors (red boxes) occur frequently.

dependencies and provides accurate semantic conditioning. This effectiveness can be attributed to its ConvNeXt-V2 (Woo et al., 2023) backbone with large convolutional kernels, which not only provide a broad receptive field but also integrate embeddings across spatial and sequential dimensions. Such integration yields more robust alignment of character content embeddings with the VAE latent space, preventing over-reliance on individual tokens. Moreover, by enriching the contextual information, the content encoder mitigates semantic ambiguity: without it, the model must rely on sparse text embeddings, which often leads to misalignment and confusion. The visual results in Figure 6 further support these observations, demonstrating that, without the content encoder, generated text exhibits poor alignment and semantic drift.

### 4.5 FINE-GRAINED CHARACTER–LAYOUT ANALYSIS

As shown in Figure 7, we visualize character centroids (red dots) for each method and connect them to reveal layout trends. Baseline methods exhibit unnatural connections between adjacent characters in certain regions (highlighted by red boxes), stemming from their assumption that character content and layout can be modeled independently. This often results in broken baselines and discontinuous structures. This phenomenon points to a fundamental property of handwritten text lines: *character layout and connections cannot be fully disentangled, as they remain inherently context-dependent.* Importantly, text-line layout is not merely arranging character bounding boxes; it should also account for the structural properties of individual characters and the contextual dependencies among their neighbors. Rather than treating layout as a separate post-processing step, DiffInk directly models the spatial arrangement and inter-character connections of the entire text line within a structure-aware latent space, thereby producing more coherent and consistent results.

| Text | Drawing | DeepIminator | WLU | SDT | OLHWG | **DiffInk (Ours)** | Target |
|---|---|---|---|---|---|---|---|
| 都系独立运作 | | | | | | | |
| 汉外长城北线 | | | | | | | |

Figure 7: **Fine-Grained Layout Analysis**. Baseline methods show local misalignments (red boxes), while DiffInk achieves more coherent and consistent layouts.

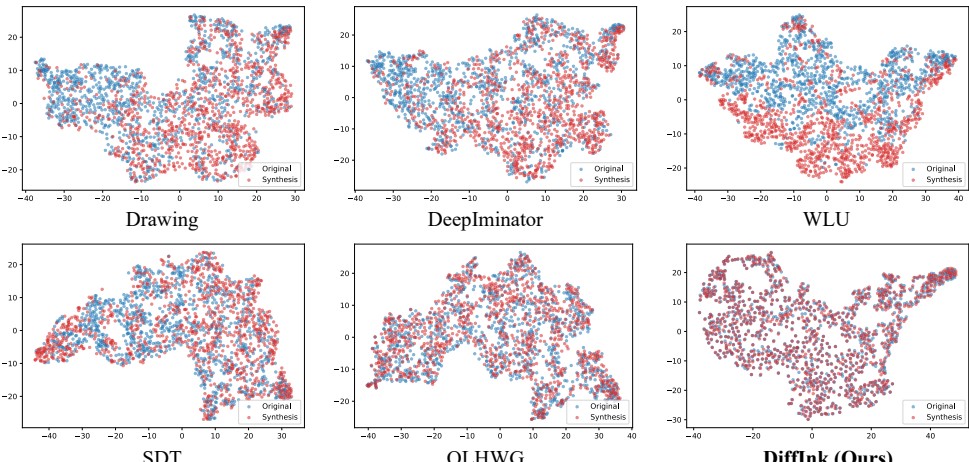

Figure 8: **Comparison of data distributions between generated and real data.** t-SNE (Maaten & Hinton, 2008) visualization of features from the same text lines generated by baselines and DiffInk. DiffInk closely overlaps with real data, while baselines show varying degrees of deviation.

### 4.6 VISUALIZATIONS WITH T-SNE

As shown in Figure 8, the t-SNE visualization of features extracted by the trajectory encoder reveals that DiffInk's generated text lines closely overlap with real handwritten samples, indicating that DiffInk effectively captures the distribution of authentic handwriting. In contrast, baseline methods that synthesize characters individually and concatenate them using bounding boxes exhibit a noticeable distributional shift from real data. This suggests that although text lines can be reconstructed through post-hoc concatenation, such pipelines struggle to model the coupling between character shapes and text-line layout, which in real handwriting arises from continuous writing dynamics and natural inter-character transitions. As a result, character-concatenation methods fail to reproduce the holistic structure of real text lines, whereas DiffInk's end-to-end modeling captures both global layout and stylistic coherence more faithfully. These results further confirm that modeling the entire text line as a unified sequence is essential for preserving realistic handwriting statistics.

## 5 CONCLUSION

In this paper, we propose **DiffInk**, a pioneering conditional latent diffusion Transformer framework for TOHG. Our key contributions are: (1) a task-aware variational autoencoder **(InkVAE)** that incorporates OCR-based and style regularization to learn a semantically structured latent space with disentangled content and style; and (2) a conditional latent diffusion Transformer **(InkDiT)** that conditions on textual input and reference styles to generate coherent pen trajectories. Built upon these components, DiffInk produces full-line handwritten text with high content accuracy, consistent style, and natural inter-character transitions. Comprehensive experiments demonstrate that DiffInk significantly outperforms existing methods, validating the effectiveness of our approach. This work establishes the effectiveness of the conditional latent diffusion model for complex online handwriting generation and highlights its potential for OCR, personalized digital handwriting applications, and human-computer interaction systems. Limitations and discussions, including extensions to paragraph-level generation and to other languages are presented in Appendix D.

### ACKNOWLEDGEMENT

This research is supported in part by the National Natural Science Foundation of China (Grant No.:62476093).

### REPRODUCIBILITY STATEMENT

Reproduction consists of dataset preparation (Sections 4.1 and Appendix A), the design of DiffInk (InkVAE and InkDiT) as described in Sections 3.3 and 3.4, implementation and training detailed in Appendix B and Section 4.2, and evaluation following Section 4.3 and Appendix B.4.

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

## APPENDIX SECTION

## A  DATASETS AND PREPROCESSING

**Dataset**  *For Chinese TOHG*, we use the CASIA-OLHWDB 2.0–2.2 (Liu et al., 2011) datasets, which contain over 50,000 online handwritten text lines collected from 1,019 writers. Due to training resource limitations, we select 500 writers based on the average length of their text lines, prioritizing those with longer and more informative content. Among them, 400 writers are used for training and 100 for testing. As shown in Figure 9, we further analyze the text-line length distribution in the training set, where the average length exceeds 30 characters (higher than the reported 26.68 characters in prior benchmarks (Yin et al., 2013)) and the maximum length goes beyond 40 characters—sufficient to cover typical text-line writing scenarios in daily life.

We observe that the character distribution in text-line data exhibits a pronounced long-tail pattern. To mitigate this issue, we design a frequency-aware data augmentation strategy.

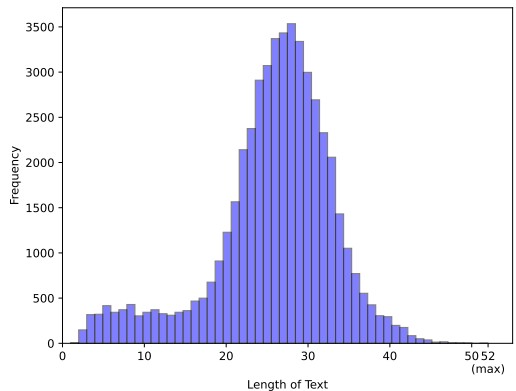

Figure 9: Analysis of text-line length distribution in the Chinese handwriting dataset. Due to dataset constraints, our method supports up to 52 characters per line, which is higher than the average of 26.68 characters reported in (Yin et al., 2013) and sufficient for practical usage.

Specifically, we sample isolated character instances written by the same author from CASIA-OLHWDB 1.0–1.2, with sampling probabilities inversely proportional to their frequencies in the original text-line corpus—assigning higher selection chances to rare characters. For each sampled author, we randomly select a text-line from the training set as a layout template, and place the sampled characters according to the original character BBoxes. This strategy enables the construction of synthetic text-lines that effectively compensate for the scarcity of low-frequency characters. In total, we generate approximately 67,000 text-line samples for training.

**Data processing** We first normalize the $(x, y)$ coordinates of all trajectory points following prior work (Zhang et al., 2017), to address scale inconsistencies caused by variations in device resolution and handwriting styles. This normalization facilitates more stable learning of structural patterns in handwritten text. Subsequently, we apply the Ramer–Douglas–Peucker(RDP) (Douglas & Peucker, 1973) algorithm to each stroke sequence to remove redundant points, thereby reducing the overall trajectory length and improving computational efficiency. The simplification coefficients are empirically selected based on extensive trial experiments: we set them to 0.4 for CASIA-OLHWDB. The average sequence length of Chinese handwriting is reduced by about 500 points after simplification. These values balance preserving handwriting structure with noise reduction, while also shortening the input sequences. For each batch of text-line data, we pad all sequences to the maximum line length. The padding regions are filled with $x = y = 0$, while the pen-state $pen$ is set to the *End of Character* symbol.

# B IMPLEMENTATION DETAILS OF DIFFINK

## B.1 DETAILS OF INKVAE

**Reconstruction and KL Divergence** As the latent modeling module of the latent diffusion Transformer, InkVAE is primarily designed to produce compact representations that minimize reconstruction loss. Specifically, InkVAE adopts a 1D convolutional encoder-decoder architecture with residual connections. The encoder processes the input handwriting sequence of shape $N \times 5$ through a series of downsampling layers, progressively transforming it into representations of size $N/2 \times 128$, $N/4 \times 256$, and finally $N/8 \times 384$, forming a latent sequence with an 8× temporal compression ratio. The decoder mirrors this structure with residual 1D convolutional blocks for upsampling, gradually reconstructing the sequence into $N/4 \times 384$, $N/2 \times 256$, and $N \times 123$. The final output of the decoder parameterizes a $p = 20$-component Gaussian Mixture Model (GMM) for $(x, y)$ trajectory prediction, with the remaining dimensions used for pen-state classification (3 classes). A Transformer-based trajectory decoder (3 layers, hidden size 256, 4 heads) is employed to enhance sequence modeling. The overall reconstruction loss $\mathcal{L}_{\text{rec}}$ consists of two components: the negative log-likelihood loss $\mathcal{L}_{\text{gmm}}$ for coordinate modeling, and the cross-entropy loss $\mathcal{L}_{\text{pen}}$ for pen-state classification. A KL divergence loss is also applied to the latent space, as is standard in VAE frameworks, to promote regularization; to balance reconstruction quality, we follow diffusion-based models and set its coefficient to a small value.

The $\mathcal{L}_{\text{gmm}}$ is defined as Equation 4, where $\mathbf{p}_t = (x_t, y_t)$ is the ground-truth point at timestep $t$, and each mixture component is parameterized by mean $\boldsymbol{\mu}_{m,t}$, covariance $\boldsymbol{\Sigma}_{m,t}$ (often diagonal), and mixing coefficient $\pi_{m,t}$. The pen-state prediction is treated as a 3-way classification task (pen-down, pen-up, end-of-stroke), with loss defined as Equation 5, where $y_{t,k}^{\text{pen}}$ is the one-hot ground-truth label and $p_{t,k}^{\text{pen}}$ is the predicted probability for class $k$ at timestep $t$. This focal loss formulation down-weights easy samples and focuses training on harder ones.

$$\mathcal{L}_{\text{gmm}} = -\frac{1}{T} \sum_{t=1}^{T} \log \left( \sum_{m=1}^{M} \pi_{m,t} \cdot \mathcal{N} \left( \mathbf{p}_t \mid \boldsymbol{\mu}_{m,t}, \boldsymbol{\Sigma}_{m,t} \right) \right), \tag{4}$$

$$\mathcal{L}_{\text{pen}}^{\text{focal}} = -\frac{1}{T} \sum_{t=1}^{T} \sum_{k=1}^{3} \alpha_k \left( 1 - p_{t,k}^{\text{pen}} \right)^{\gamma} y_{t,k}^{\text{pen}} \cdot \log p_{t,k}^{\text{pen}}, \tag{5}$$

**Task relevant Regularization** Recent studies (Tang et al., 2025; Yao et al., 2025; Guo et al., 2025) suggest that a well-structured latent space plays a crucial role in the effectiveness of downstream

---

**Algorithm 1** InkDiT Training and Inference

---

**Training Phase:**
1: **Input:** $x_0$, $x_{\text{ref}}$, $Z$
2: Sample $t \sim \mathcal{U}(1, T)$
3: Sample noise $\epsilon \sim \mathcal{N}(0, \mathbf{I})$
4: Compute $x_t = \sqrt{\bar{\alpha}_t} \cdot x_0 + \sqrt{1 - \bar{\alpha}_t} \cdot \epsilon$
5: Predict $\hat{x}_0 = \text{InkDiT}_\theta([x_t, x_{\text{ref}}, Z], t)$
6: **Update** $\text{InkDiT}_\theta$ using loss: $\mathcal{L}_{\text{diff}} = \|\hat{x}_0 - x_0\|^2$

**Inference Phase:**
1: **Input:** $x_T \sim \mathcal{N}(0, \mathbf{I})$, $x_{\text{ref}}$, $Z$
2: **for** $t = T, \ldots, 1$ **do**
3: $\quad \hat{x}_0 = \text{InkDiT}([x_t, x_{\text{ref}}, Z], t)$
4: $\quad \epsilon = (x_t - \sqrt{\bar{\alpha}_t} \cdot \hat{x}_0)/\sqrt{1 - \bar{\alpha}_t}$
5: $\quad x_{t-1} = \sqrt{\bar{\alpha}_{t-1}} \cdot \hat{x}_0 + \sqrt{1 - \bar{\alpha}_{t-1}} \cdot \epsilon$
6: **end for** $\qquad\qquad\qquad \triangleright$ Final output: $\hat{x}_0$

---

diffusion-based modeling. To this end, beyond trajectory reconstruction, InkVAE introduces two auxiliary objectives to impose task-specific regularization on the latent space: (1) A style classifier composed of an LSTM (Staudemeyer & Morris, 2019) with attention pooling and a 400-class writer identification head, and (2) a Transformer-based OCR module (hidden size 384) equipped with a 2648-class character classifier and trained with CTC loss to encourage content-aware representation learning. The style classification loss and the OCR loss are defined as Equation 6, where the former uses a cross-entropy loss between the writer identity label $y^{\text{style}}$ and the predicted probability $p^{\text{style}}$, and the latter adopts the standard CTC loss (Graves et al., 2006) to align predicted character sequences with ground-truth transcriptions.

$$\mathcal{L}_{\text{sty}} = -\sum_{c=1}^{C} y_c^{\text{style}} \cdot \log p_c^{\text{style}}, \quad \mathcal{L}_{\text{ocr}} = \text{CTC}(\mathbf{P}^{\text{ocr}}, \mathbf{y}^{\text{ocr}}). \tag{6}$$

### B.2 DETAILS OF INKDIT

To generate text lines conditioned on character sequences, InkDiT takes as input three components: the noisy latent representation $x_t$, a style condition $x_{\text{ref}}$, and the corresponding text embeddings. The text embeddings is processed by a lightweight encoder to produce content-aware features. These three inputs are then fused and projected into a unified joint space, which is subsequently modeled by a multi-layer Transformer for time-aware latent denoising.

**Content Encoder**    To model the character-level semantics in handwriting generation, we employ a lightweight content encoder based on three stacked ConvNeXt-V2 (Woo et al., 2023) blocks. The input to this encoder is a sequence of padded character embeddings with a dimensionality of 512. Each ConvNeXt-V2 block consists of a depthwise 1D convolution with kernel size 7 to capture local context, followed by a LayerNorm layer and a two-layer feedforward subnetwork with GELU activation and Global Response Normalization (GRN). Residual connections are applied to stabilize training and preserve lower-level features. This architectural design enables the encoder to effectively extract local and contextual dependencies among character tokens, resulting in content-aware features that are well aligned with the latent space representation used by the diffusion model.

**DiT Backbone**    The InkDiT backbone operates entirely within a 384-dimensional latent space. During each denoising step, it receives three conditional inputs: the noised latent representation $x_t \in \mathbb{R}^{l \times 384}$, a style condition feature $x_{\text{ref}} \in \mathbb{R}^{l \times 384}$, and the content feature derived from the ConvNeXt-based encoder. These inputs are fused through a mixing operation and subsequently projected to a 896-dimensional joint space using a linear layer. The resulting fused representation is then passed into a 16-layer Transformer model that follows the standard design of multi-head self-attention and feedforward networks. Each Transformer block incorporates adaptive normalization and gated residual connections modulated by timestep embeddings, enabling the model to perform time-aware denoising. This design facilitates joint reasoning over content semantics, spatial layout, and temporal dynamics within a unified generative framework.

**Diffusion Process**    The diffusion process follows a cosine noise schedule with $T = 1000$ timesteps. Instead of predicting the noise, InkDiT is trained to directly estimate the denoised latent variable $x_0$, following the DDPM-style parameterization. During inference, DDIM (Song et al., 2020) sampling with 5 steps is used to efficiently generate the final latent output.

The complete training and inference procedures are summarized in Algorithm 1. During training, the model learns to recover $x_0$ from its noisy version $x_t$, conditioned on both textual input and reference

trajectory. To further improve generation quality, we apply lightweight DDIM-specific fine-tuning, where the model is supervised to predict $\hat{x}_0$ at intermediate steps by unrolling a few DDIM iterations. This enhances fidelity with minimal computational overhead. In inference, generation starts from Gaussian noise at $t = T$ and iteratively denoises via DDIM updates, producing a coherent and stylistically aligned latent sequence that is ultimately decoded into an online handwriting trajectory.

## B.3 TRAINING CONFIGURATION

For InkVAE, all components—including the sequence encoder, sequence decoder, and two regularization classifiers—are jointly optimized with a learning rate of $5 \times 10^{-4}$, $\beta = (0.9, 0.99)$, and a weight decay of $1 \times 10^{-4}$. To stabilize training, we apply gradient clipping with a maximum norm of 5.0. For InkDiT, we adopt the same settings but use a smaller learning rate of $7.5 \times 10^{-5}$ and a stricter gradient clipping norm of 1.0. In both cases, the learning rate schedule combines a 5% linear warm-up phase followed by cosine decay, implemented using the `LambdaLR` scheduler in PyTorch. All models are trained from scratch on 4×NVIDIA RTX A6000 GPUs. Training InkVAE takes about 16 hours, whereas InkDiT training requires roughly 3 days. We employ the AdamW optimizer (Loshchilov & Hutter, 2017) for both stages of training.

## B.4 EVALUATION METRICS

To evaluate the readability and correctness of generated text lines, we adopt Accurate Rate (AR) and Correct Rate (CR) as defined in Equations 7 and 8. Following (Liu et al., 2011), $N_t$ is the total number of ground-truth characters; $D_e$, $S_e$, and $I_e$ denote the numbers of deletion, substitution, and insertion errors, respectively. Specifically, we train a text-line OCR recognition model solely on the training set, which still achieves a high accuracy of 97.62% when directly evaluated on the test set. The model adopts the encoder of InkVAE as the backbone, followed by a character classification head, and is subsequently used to evaluate the textual accuracy of generated handwriting samples.

$$\text{AR} = \frac{(N_t - D_e - S_e - I_e)}{N_t} \times 100\%$$

(7)

$$\text{CR} = \frac{(N_t - D_e - S_e)}{N_t} \times 100\%$$

(8)

The text-line DTW (Berndt & Clifford, 1994) distance is computed over the $(x, y)$ coordinate sequences of each full text line and normalized by the length of the corresponding ground-truth trajectory, following the formulation in Equation 9. Here, $N$ is the number of test samples, $|x_i^{\text{gt}}|$ denotes the length of the $i$-th ground-truth trajectory, and $\text{DTW}(\cdot)$ is efficiently computed using the `torch-fastdtw` library.

$$\text{Norm-DTW} = \frac{\text{DTW}(x_i^{\text{gt}}, x_i^{\text{pred}})}{|x_i^{\text{gt}}|}$$

(9)

The style score is obtained by feeding the generated text lines into a style classifier, predicting their corresponding writer identities. To this end, we train a 100-class writer identification model on the testing set, which achieves a classification accuracy of 99.9% on the test set, demonstrating its strong discriminative ability for style evaluation. The style classifier consists of the encoder of InkVAE followed by a writer classification head. Besides, generation efficiency (Avg. char/s) is evaluated as the average number of characters generated per second. All methods are tested with a batch size of 1 on the same NVIDIA RTX A6000 GPU. The average generation speed is computed by dividing the total number of generated characters by the elapsed time, excluding data loading and other preprocessing operations.

| | | | |
|---|---|---|---|
| **Writer-1** | 他同时也是一位非常度诚的基督徒 | 的急剧著缩，是困找长江中下游地区的 | 政府成立了成吉思汗陵管理局，也尔庐 |
| **Writer-2** | 他同时也是一位非常度城的基督徒 | 的急剧著缩，是困扰色江中下游区域的 | 政府成立了成吉思汗陵管理局，达尔庐 |
| **Writer-3** | 他同时也是一位非常度诚的基督徒 | 的急剧著缩，是困扰长江中下游之区 | 政府成立了成吉思汗陵管理局，达尔庐 |
| **Writer-4** | 他门司时也是一位非常度减的基督徒 | 的急剧著缩，是困扰江中扑游区域的 | 政府成立了成古思汗管理局，达东庐 |
| **Text Line** | "他同时也是一位非常度诚的基督徒" | "的极具萎缩，是困扰长江中下游地区的" | "政府成立了成吉思汗管理局，达尔庐" |

Figure 10: **Same text across different styles.** Generating the same text in different styles shows that DiffInk captures distinct style variations while maintaining intra-writer consistency.

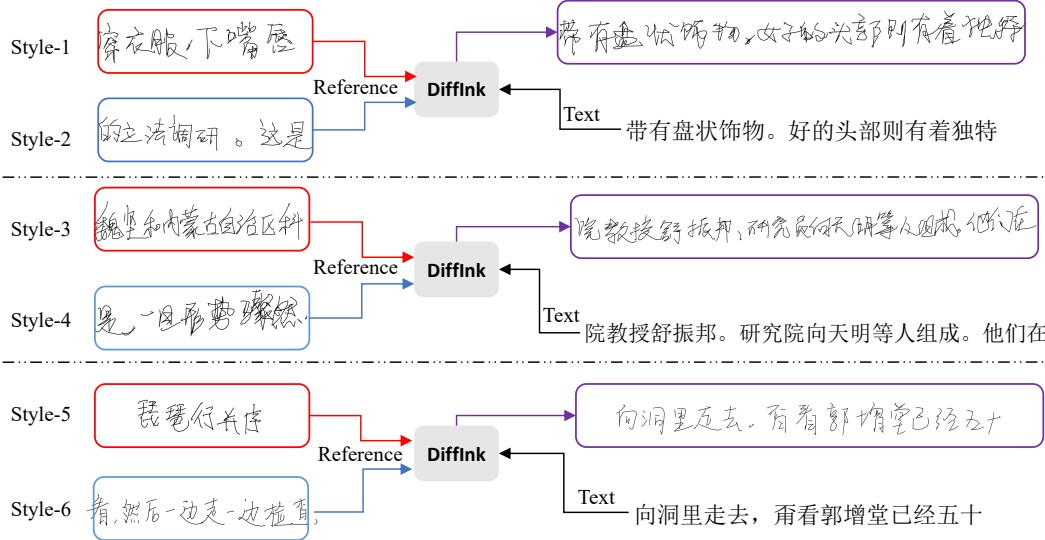

Figure 11: **Multi-style input.** DiffInk can take two style references simultaneously and, conditioned on the given text, generate handwriting that blends characteristics of both styles, enabling controllable style manipulation.

## C  MORE EXPERIMENTAL RESULTS

### C.1  STYLE CONTROL AND CONSISTENCY

**Style Consistency**  As illustrated in the Figure 10, we showcase generated text lines from four writers in the test set. Each writer is conditioned on the same textual content to generate handwriting samples in their respective styles. Horizontally, despite the variation in content, the generated samples from the same writer exhibit strong stylistic consistency. For example, writer 1 consistently produces slanted layouts from the lower left to the upper right. Vertically, given the same content input, samples generated in different styles show clear variations in both layout and writing traits, such as overall length, character size, and spatial arrangement.

**Style Control**  DiffInk is capable of handling style-mixing inputs, allowing the model to generate handwriting that interpolates between multiple reference styles. As illustrated in the Figure 11, given two style examples and a target text string, the model synthesizes pen-trajectory sequences that simultaneously reflect structural consistency with the textual content and stylistic characteristics drawn from both references.

### C.2  MORE COMPARISONS WITH EXISTING METHODS

We provide additional comparisons with existing approaches, including Drawing (Zhang et al., 2017), Deepiminator (Zhao et al., 2020), WLU (Tang & Lian, 2021), SDT (Dai et al., 2023), and

OLHWG (Ren et al., 2025). As illustrated in Figure 12 and Figure 13, our method delivers superior text-line generation quality over a broader spectrum of handwriting styles.

**Writer Style 1**

| Text | 涌入黄河,从小西湖大桥附近开始,黄河南岸的半幅河面形成了一个 | 北京市交管局提供的一组数字显示,今年1月至5月,12岁以下儿 |
|---|---|---|
| Drawing | | |
| Deep Iminator | | |
| WLU | | |
| SDT | | |
| OLHWG | | |
| **DiffInk (Ours)** | | |
| Target | | |

**Writer Style 2**

| Text | 袁力表示,根据实际情况,可能会对基础费率做出调整。而根据《条例》 | Cerberus资本管理公司出资74亿美元收购未来克莱斯勒80.1%股权 |
|---|---|---|
| Drawing | | |
| Deep Iminator | | |
| WLU | | |
| SDT | | |
| OLHWG | | |
| **DiffInk (Ours)** | | |
| Target | | |

**Writer Style 3**

| Text | 据美国《防务新闻》网站报道,日前,负责亚太事务的美国防部副 | 日本三得利公司日前宣布开发出一种利用植物净化水质的新方法, |
|---|---|---|
| Drawing | | |
| Deep Iminator | | |
| WLU | | |
| SDT | | |
| OLHWG | | |
| **DiffInk (Ours)** | | |
| Target | | |

Figure 12: **More visual comparisons with SOTA methods.** DiffInk generates more natural text-line results, while the red boxes highlight unnatural character connections.

**Writer Style 4**

| Text | 节能减排力度, 力争年内实现主要污染物二氧化硫比上年减排20% | 农业部副部长牛盾8日表示, 当前在经济全球化大背景下, 我国棉花产业 |
| --- | --- | --- |
| Drawing | | |
| Deep Iminator | | |
| WLU | | |
| SDT | | |
| OLHWG | | |
| **DiffInk (Ours)** | | |
| Target | | |

**Writer Style 5**

| Text | 盖保罗1996年被任命为欧莱雅(中国)有限公司总裁。他通过 | "硫酸根吸附剂及其处理工艺可以有效解决我国煤矿 |
| --- | --- | --- |
| Drawing | | |
| Deep Iminator | | |
| WLU | | |
| SDT | | |
| OLHWG | | |
| **DiffInk (Ours)** | | |
| Target | | |

**Writer Style 6**

| Text | 量大约比冥王星大27%, 是目前已知最大的矮行 | 首先登场的是宝瓶座厄塔流星雨, 5月6日流星雨极盛, 为正在 |
| --- | --- | --- |
| Drawing | | |
| Deep Iminator | | |
| WLU | | |
| SDT | | |
| OLHWG | | |
| **DiffInk (Ours)** | | |
| Target | | |

Figure 13: **More visual comparisons with SOTA methods.** DiffInk generates more natural text-line results, while the red boxes highlight unnatural character connections.

C.3 MORE RESULTS GENERATED BY DIFFINK

In Figure 14, we present additional results generated by DiffInk. We randomly sample eight writers from the test set and generate multiple text lines for each of them. Visual inspection reveals that the generated text lines from the same writer exhibit consistent local stroke patterns and global layout styles, while clear stylistic differences emerge across different writers. These results demonstrate that DiffInk effectively captures inter-writer style variation, and that our lightweight style classification strategy improves the model's ability to generalize across styles. Moreover, the generated text lines also maintain high content accuracy and readability, further validating the effectiveness of the proposed DiffInk framework.

### Writer Style 1

Text: 的呆坏账增多等一系列连锁反应,从而影响经济和金融的

Text: 管理论坛"更名为"世界经济论坛",也称达沃斯论

Text: 这是一个经常被提及的话题,但在实践中,这又是一个未及

Text: 《条例》规定,调整保险费率的幅度较大的,保监会应

### Writer Style 2

Text: 但是,相较传统的纸质阅读,通过网络及电视等媒体进行

Text: 经营、管理上存在什么问题,最终的结局如果是走进监狱,

Text: 现今不到短短三个月的时间,消息就已变成现实,这样的高效率

Text: 罪被捕入狱的人数不断增多,此方面的报道也屡屡见诸报端。

### Writer Style 3

Text: 5月的苍穹好戏连台,公众不但可以观赏到宝瓶座厄塔流星雨

Text: 62亿年前就已存在,比科学家此前认为的早5亿年。

Text: 的要强得多。这就是说,磁气圈当时强到足以保护地球免

Text: 基因库,于1992年列入《国际重要湿地名录》。保护区内水生植被当丰富.大

### Writer Style 4

Text: 出。信知生男恶,反是生女好。生女犹得嫁比邻,生男埋

Text: 同是天涯沦落人,相逢何必曾相识。我从去年辞帝京,谪居卧病浔

Text: 梦少年事,梦啼妆泪红阑干。我闻琵琶已叹息,又闻此语重唧唧

Text: 然。自叙少小时欢乐事,今漂沦憔悴,转徙于江湖间。予出官二年,

### Writer Style 5

Text: 湖南时提出,要将洞庭湖主湖面恢复到解放初期

Text: 理学家已进行了地震和地磁测量,以探测这块鲜为人知的

Text: 期,第272窟就是这一时期的代表作。记者从敦煌研究院了

Text: 这块海底大陆被称为克尔格伦高原,大约约等于德国

### Writer Style 6

Text: 赵高是第一流的书法家、文字学家,也是精通法律的专

Text: 他体魄高大强壮,骑术车技精湛,武艺非同寻常,是秦帝国宫

Text: 23号演习实兵验证阶段将于5月14日起展开。值得注意的是

Text: 兰进行海空联合截击的精密武器射击;中部将在台中港与

### Writer Style 7

Text: 的重大任务。我们要从贯彻落实科学发展观、构建社会主义和谐

Text: 年非洲开发银行集团理事会年会开幕式,并发表讲话

Text:有骨质疏松家族史者易患骨质疏松症。对骨峰值的建立,遗传因

Text:摩过程中也会适当地激活血液的流通,刺激神经末梢,

### Writer Style 8

Text: 是夕始觉有迁谪意。因为长句,歌以赠之,凡六百一十六言,命曰《琵琶行》

Text:京城女,家在虾蟆陵下住。十三学得琵琶成,名属教坊第一部。

Text:流雪山。好为庐山谣,兴因庐山发。闲窥石镜清我心,谢公行

Text:平生不得意。低眉信手续续弹,说尽心中无限事。轻拢慢捻抹

Figure 14: **More generation results by DiffInk (Ours)**. DiffInk is capable of producing diverse handwritten long text lines, exhibiting high consistency within the same style and clear distinctions across different styles.

Table 4: **DDIM sampling–step analysis of DiffInk**, illustrating the trade-off between generation quality and inference speed.

| Sample Steps | AR% ↑ | CR% ↑ | Style ↑ | DTW ↓ | Time Cost |
|---|---|---|---|---|---|
| 1 | 36.23 | 36.39 | 35.85 | 1.028 | 0.2x |
| 2 | 79.70 | 80.00 | 66.31 | 1.022 | 0.4x |
| 3 | 89.93 | 90.19 | 73.51 | 1.026 | 0.6x |
| 4 | 93.16 | 93.39 | 76.47 | 1.040 | 0.8 |
| **5 (Ours)** | **94.38** | **94.58** | **77.38** | **1.052** | **1x** |
| 6 | 94.86 | 95.01 | 77.69 | 1.058 | 1.2x |
| 10 | 95.35 | 95.52 | 78.32 | 1.067 | 2.0x |
| 15 | 95.36 | 95.56 | 79.10 | 1.068 | 3.0x |
| 20 | 95.27 | 95.47 | 79.16 | 1.076 | 4.0x |

### C.4 DDIM SAMPLING STEPS EVALUATION

To evaluate the impact of DDIM sampling steps on generation performance, we conducted an additional experiment. As shown in the table 4, We observe that the performance of DDIM sampling continues to improve as the number of steps increases from 1 to 5. However, when the number of steps exceeds 10, the gains in content and style scores become marginal, while the DTW value increases and the sampling cost grows significantly. Therefore, we adopt 5-step DDIM sampling in this paper as a practical trade-off between generation quality and computational efficiency.

## D DISCUSSION

### D.1 COMPARISON WITH TWO-STAGE METHODS

As illustrated in Figure 15, conventional two-stage methods decouple character generation from layout prediction and rely on additional post-processing to stitch characters together, which often results in less coherent text lines. In contrast, our proposed DiffInk models content, style, and layout within a unified framework, enabling genuine end-to-end generation that produces text lines with more natural character connections and globally consistent layouts.

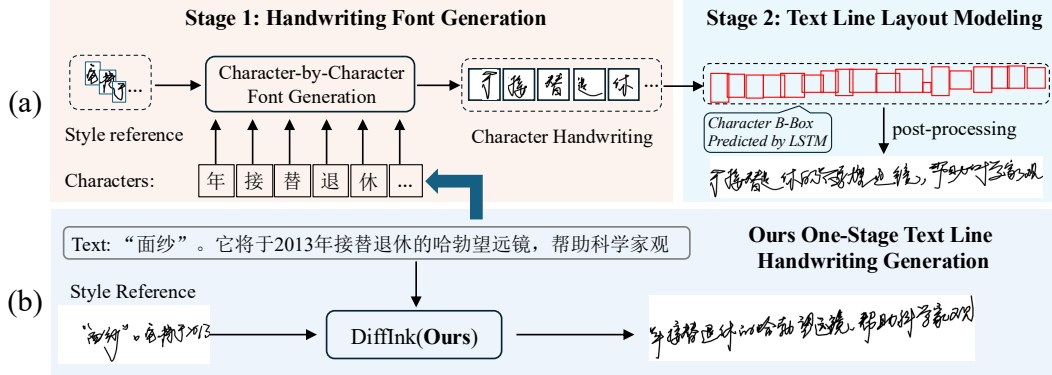

Figure 15: **Comparison with character–layout decoupled approaches**: (a) a two-stage pipeline combining handwritten font generation with layout post-processing; (b) DiffInk, which takes text and a style reference to directly output complete text lines. Unlike the two-stage pipeline, DiffInk generates more natural character connections rather than mechanically stitching bounding boxes.

### D.2 IMPLEMENTATION OF DIFFINK

In the main experiments, we use a continuous reference segment from each text line as the style reference, both for convenient sampling and to evaluate whether the model can generate layouts that follow the trend of this given reference. This setting is adopted for convenient comparison with

Figure 16: **Results on the IAM test set.** DiffInk generates English text lines with consistent style and accurate content. The top row shows generated results, and the bottom row shows the corresponding target samples.

Table 5: Recognition performance on ICDAR13 (Yin et al., 2013) with and without our synthesized data. Adding synthetic text lines improves AR and CR by 8.5 and 8.6 points, demonstrating both the effectiveness and quality of the generated data.

| OCR Model | Training Data | CR ↑ | AR ↑ |
|---|---|---|---|
| CNN+Transformer | Real data only (Baseline) | 83.9 | 84.2 |
| | **+ DiffInk Synth (Ours)** | **89.6** | **90.0** |
| GLRNet (Peng et al., 2021) | Real data only (Baseline) | 86.2 | 86.6 |
| | **+ DiffInk Synth (Ours)** | **90.9** | **91.1** |

recent SOTA methods (Ren et al., 2025), which feed the layouts of the first ten characters of a real text line into the model; the model then predicts the positions and sizes of subsequent characters based on the reference layout, the input character content, and an average character-size prior. To ensure a fair comparison, we instead provide the continuous handwritten trajectory segment of the real text line—containing both style and layout information—as input to our model. For the other compared methods, the same reference segment is used as the style condition. The corresponding character bounding boxes of this reference are fed into the OLHWG layout prediction module as prompts, and the final text-line results are obtained by scaling the generated characters according to the predicted bounding boxes. Finally, line-level metrics—including content scores, style scores, and DTW distance—are computed with each text line conditioned on its own reference segment.

### D.3 APPLYING TO OTHER LANGUAGES

To validate the generalizability of the proposed method, we directly applied it to English text line generation tasks and conducted experiments on the IAM-OnDB dataset. The data processing pipeline remained consistent with the Chinese setup, including the use of the RDP algorithm (with the redundancy removal parameter set to 0.5) for redundant point removal and normalization of trajectory coordinates. The IAM-OnDB (Liwicki & Bunke, 2005) dataset contains approximately 10,000 handwritten English text line samples produced by 221 writers, featuring substantial cursive writing. We randomly selected data from 25 writers as the test set, with the remaining data used for training. During training, all hyperparameters remained consistent with the Chinese experimental configuration, except for the learning rate, which was reduced to $5 \times 10^{-5}$.

Figure 16 showcases several examples of DiffInk applied to English handwriting generation. The model effectively generates text-line sequences that are both content-accurate and style-consistent, even when the handwriting features strong cursive connections and diverse stylistic variations. These results highlight the versatility and generalizability of the DiffInk framework across multilingual and stylistically complex handwriting scenarios.

### D.4 IMPROVING OCR PERFORMANCE WITH SYNTHETIC DATA

In practical applications, the quantity of handwriting data plays a critical role in improving OCR performance. To assess the impact of our synthesized data, we use the generated text lines to augment the original training set. For recognition, we adopt the same OCR model employed in this work for evaluation—a CNN+Transformer architecture—as the text-line recognition backbone. In addition, we also follow the recognition model design used in GLRNet (Peng et al., 2021) and conduct experiments under the same settings.

Our experimental results on the ICDAR13 dataset (Yin et al., 2013) are presented in Table 5. It can be observed that the proposed generation method significantly improves the performance of the recognition model. Specifically, after augmenting the training set with our synthesized data, the AR and CR metrics increased by 8.5 and 8.6 percentage points, respectively. When using the recognition model from GLRNet (Peng et al., 2021), our synthesized data further yields an average improvement of around 4 percentage points. This demonstrates that our synthesized data can effectively mitigate recognition degradation caused by the long-tail distribution of characters and the limited diversity of handwriting styles, thereby substantially enhancing the representational capacity of existing datasets and further confirming the high content and style quality of our synthetic data.

### D.5 PERSONALIZED HANDWRITING GENERATION WITH DIFFINK

As a text-line handwriting generation model, **DiffInk (Ours)** is useful not only for producing high-quality synthetic data for OCR systems, but also for broader generative applications such as personalized handwriting services and human–computer interaction. For example, in a personalized handwriting generation scenario: for a given writer, only a single reference line—or even a short trajectory of fewer than ten characters—is provided as the prompt $X_{\text{ref}}$, with its corresponding text $T_{\text{ref}}$. The user then specifies a target text line $T_{\text{gen}}$, and the model generates the corresponding trajectory $X_{\text{gen}}$. As shown in Figure 17, DiffInk yields text lines with coherent layouts under this one-shot setting. To further enhance diversity, the reference trajectory $X_{\text{ref}}$ can be augmented to simulate different structural layouts. Furthermore, our approach offers a promising direction for paragraph-level generation. As illustrated in Figure 18, we present an example of a model-generated handwritten paragraph of classical Chinese poetry. By simply concatenating the line-level results, a complete paragraph can be obtained, further demonstrating that DiffInk can support a broad range of handwriting-related applications.

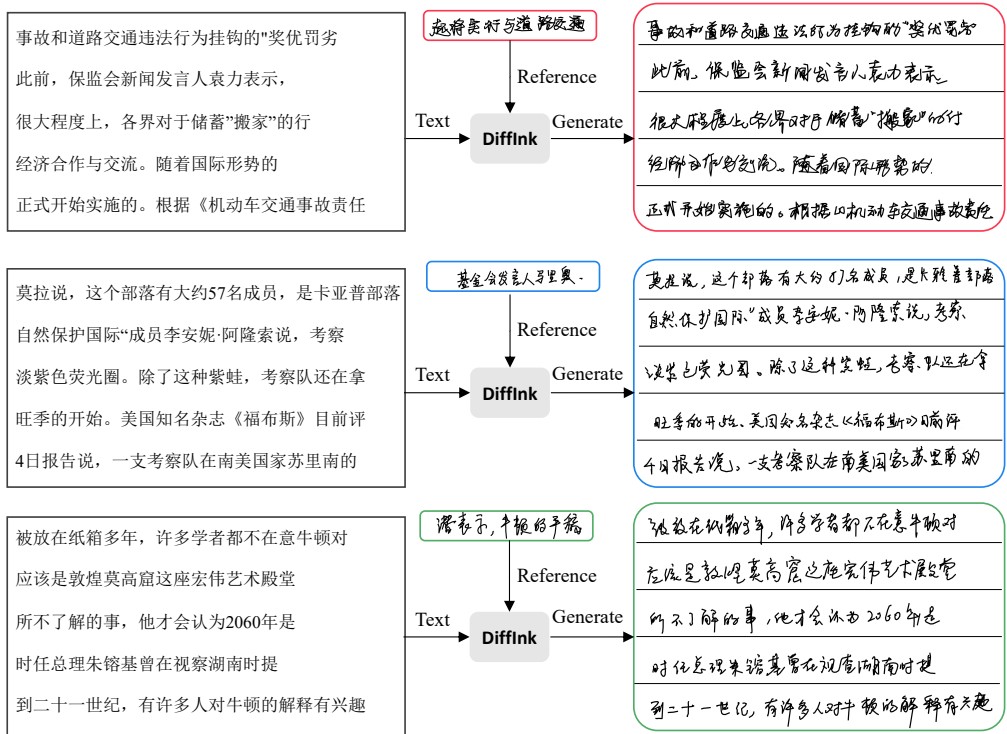

Figure 17: **One-shot generation results**. DiffInk is capable of producing text lines of arbitrary length conditioned only on a short reference trajectory, supporting arbitrary character combinations from the defined vocabulary. Even under this one-shot setting, the generated lines maintain both high character accuracy and consistent writing style. Different colors denote different styles.

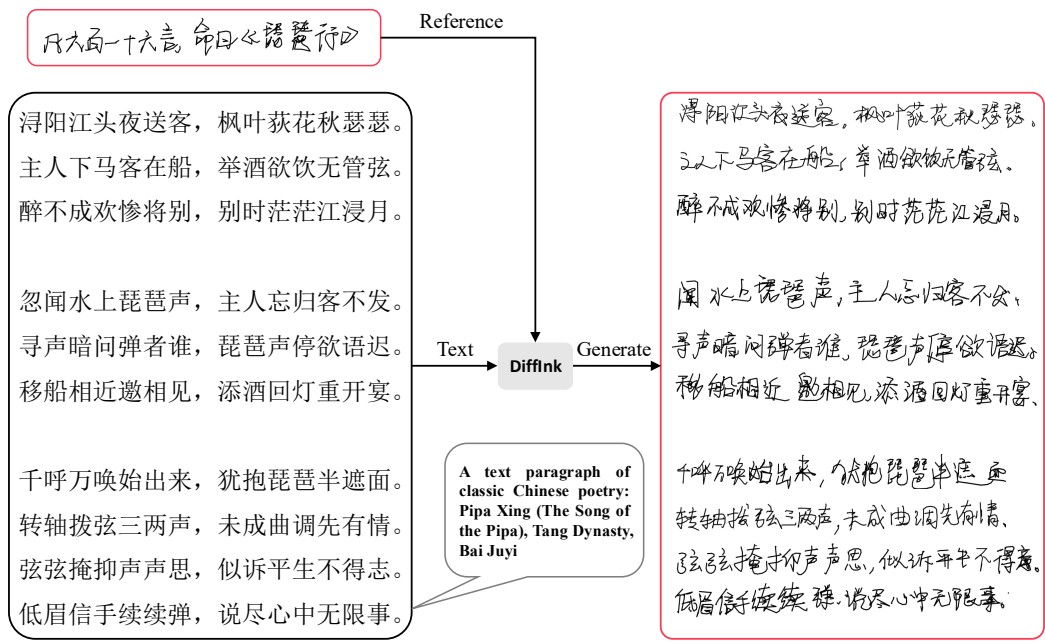

Figure 18: **Extending handwriting generation to the paragraph level**, we take classical Chinese poetry as an example. Given a style reference, the text is input line by line to produce line-level results, which are then concatenated to form a complete handwritten paragraph.

### D.6 LIMITATIONS

While our model establishes SOTA performance on the notoriously difficult task of Chinese hand-written text-line generation, direct application to other languages (e.g., Latin-based scripts) still requires retraining. Exploring how to develop a unified multilingual generation model with only minor modifications will therefore be one of our key directions for future work.

## E USE OF LARGE LANGUAGE MODELS (LLMs)

LLMs, such as ChatGPT (OpenAI, 2023) and DeepSeek (DeepSeek-AI, 2024), were employed solely for language polishing and minor writing refinements in this paper. They were not used for generating ideas, designing methods, conducting experiments, analyzing results, or creating figures. All scientific contributions, including the conception of the research problem, model development, and experimental validation, were carried out independently by the authors.

