# OpenReview forum: "DiffInk: Glyph- and Style-Aware Latent Diffusion Transformer for Text to Online Handwriting Generation"
_ICLR.cc/2026/Conference — ICLR 2026 Poster_

### Official Review · Reviewer_7DmP · 2025-10-27

**Soundness:** 2
**Presentation:** 3
**Contribution:** 2
**Rating:** 4
**Confidence:** 4

**Summary:**

The paper introduces DiffInk, a framework for text-to-online handwriting generation (TOHG) that generates full-line pen trajectories conditioned on input text and a reference handwriting style. DiffInk is based on latent diffusion, consists of two core components: InkVAE and InkDIT.

**Strengths:**

1.  The application task is rare yet interesting, yielding strong results in both visual and quantitative evaluations.

2. The method is clearly presented and easy to follow.

3. Each sub-module’s motivation is well justified, and ablation studies are provided.

**Weaknesses:**

1.  Although commendable performance has been achieved on the specific task, the overall framework more closely resembles an assemblage of existing techniques, limiting its degree of novelty. Specifically, reference [1] already employs a latent diffusion model to generate handwritten text lines, the present work differs mainly in the data format (offline or online) and the way how textual content is encoded. As for the content-encoding strategy, reference [2] likewise adopts a character embedding combined with a lightweight content encoder.

2. There are relatively few baselines, partly because prior work on full-line text generation is scarce. However, given the abundance of baselines for single-character generation, I believe this task can be viewed as a special case of line-level generation and constitutes a valid evaluation setting for assessing fine-grained handwriting style imitation.

3. Several implementation details in the experiment section can be stated more clearly, please refer to the questions (as well as some issues).



----------

[1]. Template-guided cascaded diffusion for stylized handwritten Chinese text-line generation.

[2]. Content and style aware generation of text-line images for handwriting recognition.

**Questions:**

1. Regarding the evaluation metrics, the handwritten-line recognition model and the writer-identification model both employ the encoder of the proposed InkVAE as their backbone. Is this the exact network obtained during the training of InkVAE, or was an additional network trained separately?

2. What loss function is used for the writer-identification model? Moreover, given that the training and test sets correspond to disjoint writer classes, why is it valid to perform inference on the test set after training solely on the training set?

3. The method employs data augmentation to enlarge the training set by selecting individual characters from a single character dataset and inserting them into the existing line layouts. Since the synthesized characters appear disconnected, this disrupts the connections, seemingly contradicting the argument made in Section 4.5. What would the performance be if this form of data augmentation were removed?

4. The comparison of generation speeds needs further clarification. For each method, what batch size was used? Moreover, for the prior diffusion based approaches, does the observed acceleration stem mainly from DDIM’s faster sampling schedule or from the lower dimensionality of the latent space? In addition, what is the relationship between the number of DDIM inference steps and the resulting generation quality?

**Details Of Ethics Concerns:**

NA.

---

> ### Author Response · Authors · 2025-11-24
> **Response to Reviewer 7DmP [1/4]**
>
> We thank the reviewer for the thoughtful comments and the valuable feedback. We are glad that the reviewer finds the task interesting, and appreciates the clarity of our method description, the ease of reproducibility, the sound design of each submodule, and the thoroughness of our ablation studies.
>
> We address the reviewer’s concerns and questions as follows.
>
> ---
>
> - *W1: Regarding the concern that the overall framework appears to be a combination of existing techniques.*
>
> **Use of latent diffusion models.**
>
> DiffInk differs substantially from [1] in **both modeling paradigm and technical implementation.** In [1], content and layout are decoupled, and two U-Net–based diffusion models are trained in separate stages: (1) one model predicts the character-level bounding boxes along a text line, and (2) another model stylizes the text-line template using the predicted character bounding boxes. The key point is that, the method in [1] employs **two separate latent diffusion models to guide content and style generation independently**, making the overall pipeline relatively complex and difficult to follow.
>
> In contrast, we proposes a novel modeling paradigm: an **end-to-end handwritten text-line generator that models both content and style jointly within a unified latent diffusion Transformer.** The design is conceptually simple, easy to understand and reproduce, and demonstrates strong empirical performance. Beyond directly applying latent diffusion, we introduce InkVAE—featuring glyph-aware and style-aware regularization—to address the core challenges of content accuracy and style consistency in long-sequence handwriting generation. Extensive ablations verify the effectiveness of this design.
>
> We would like to clarify that the publication date of this paper[1] is very close to ICLR’s submission deadline, so it was not possible for DiffInk to adopt its technical designs. In addition, the two methods differ significantly in both motivation and implementation details.
>
> **Similarity in text encoding strategies.**
>
> Using embedding-based textual representations to guide handwriting generation has been widely validated as an effective strategy in prior work [3][4][5]. DiffInk also adopts an embedding-based representation for textual content; however, its application scenario and underlying design philosophy differ substantially from those of method [2] referenced by the reviewer.
>
> Although both methods represent textual content using embeddings, their alignment designs and objectives are fundamentally different. We think that, in our scenario, the proposed approach is more flexible, and the ablation studies further demonstrate its effectiveness.
> In [2], the embedding of each character is repeated multiple times to align its temporal dimension with the style-related tokens. In [2], the embeddings of characters are **simply repeated to align with the style features, and a feature fusion mechanism** is then applied to generate the corresponding handwritten results. In our framework, we instead pad the character embedding sequence **to a unified dimensionality and employ a lightweight content encoder** based on ConvNeXtV2 to provide content feature conditions for the diffusion model, guiding it to generate the corresponding results.
>
> **Summary.**
>
> Overall, DiffInk introduces the first end-to-end framework for handwritten text-line generation built upon a general latent diffusion model. The design is novel, coherent, and well motivated, providing a general framework that achieves state-of-the-art performance on the highly challenging task of complex Chinese handwriting synthesis.We believe that DiffInk makes meaningful advances in both modeling paradigm and key technical components.
>
> ---
>
> References:
>
> [1] Template-Guided Cascaded Diﬀusion for Stylized Handwritten Chinese Text-Line Generation. In ICDAR 2025.
>
> [2] Content and Style Aware Generation of Text-line Images for Handwriting Recognition. In TPAMI 2021.
>
> [3] Ren M S, Zhang Y M, Chen Y. Decoupling layout from glyph in online Chinese handwriting generation. In ICLR 2025.
>
> [4] Ren, Min-Si, et al. Diff-writer: A diffusion model-based stylized online handwritten chinese character generator.In ICNIP 2023.
>
> [5] Alonso E, Moysset B, Messina R. Adversarial generation of handwritten text images conditioned on sequences. In ICDAR 2019.

---

> ### Author Response · Authors · 2025-11-24
> **Response to Reviewer 7DmP [2/4]**
>
> - *W2: relatively few baselines.*
>
> Thank you for the suggestion. In the revised version, we have **added two additional baselines**—Drawing [1] and Deep-Imitator [2]—both derived from the SketchRNN[3] and widely used for handwritten character generation. The detailed experimental results are shown in the table below, and the superiority of our method remains clearly demonstrated. We believe these experiments have made our paper more convincing, and we would appreciate any additional comparison methods the reviewers may suggest.
>
> ### Quantitative comparison with SOTA handwriting generation methods.
>
> | **Method** | **Venue** | **Output** | **AR% ↑** | **CR% ↑** | **Style ↑** | **DTW ↓** | **Avg. char/s ↑** |
> |------------|-----------|------------|-----------|-----------|-------------|-----------|-------------------|
> | **Drawing (New added)**       | TPAMI'17 | Char-level | 76.35 | 76.87 | 25.75 | 1.582 | 4.16 |
> | **DeepImitator (New added)**  | PR'20 | Char-level | 78.21 | 78.98 | 27.62 | 1.561 | 3.96 |
> | WLU           | CGF'21 | Char-level | 79.85 | 80.23 | 35.71 | 1.540 | 25.19 |
> | SDT           | CVPR'23 | Char-level | 82.53 | 83.00 | 50.51 | 1.270 | 3.35 |
> | OLHWG         | ICLR'25 | Char+layout | 91.48 | 91.71 | 44.74 | 1.326 | 0.07 |
> | **DiffInk (Ours)** | **This work** | **Line-level** | **94.38** | **94.58** | **77.38** | **1.049** | **58.47** |
>
> ---
>
> - *Q1&Q2: Regarding the reviewer’s concerns about the evaluation metrics.*
>
> The OCR model used for evaluation is **trained independently** and is architecturally consistent with the sequence encoder and OCR regularization head in InkVAE; this was stated in the appendix, and we have now highlighted it more clearly in the main paper. The writer-classification model is trained on the **test set using a standard classification loss (Cross-Entropy Loss).** We apologize for the earlier typo where it was mistakenly described as being trained on the training set.
>
> The construction of our evaluation metrics follows established practices in prior work, including SDT [4], FontRNN [5], and the ICDAR 2013 Competition [6].
>
> ---
>
> References:
>
> [1] Zhang X Y, Yin F, Zhang Y M, et al. Drawing and recognizing chinese characters with recurrent neural network. In TPAMI 2017.
>
> [2] Zhao B, Tao J, Yang M, et al. Deep imitator: Handwriting calligraphy imitation via deep attention networks. In Pattern Recognition 2020.
>
> [3] Ha D, Eck D. A neural representation of sketch drawings. arXiv preprint arXiv:1704.03477, 2017.
>
> [4] Dai G, Zhang Y, Wang Q, et al. Disentangling writer and character styles for handwriting generation. In CVPR 2023.
>
> [5] Tang S, Xia Z, Lian Z, et al. FontRNN: Generating Large‐scale Chinese Fonts via Recurrent Neural Network. In Computer Graphics Forum 2019.
>
> [6] Fei Yin, Qiu-Feng Wang, Xu-Yao Zhang, and Cheng-Lin Liu. In ICDAR 2013.

---

> ### Author Response · Authors · 2025-11-24
> **Response to Reviewer 7DmP [3/4]**
>
> - *Q3: Regarding the reviewer’s concern about the use of synthetic data.*
>
> **First, we would like to emphasize that the use of synthetic data as a data augmentation strategy is widely adopted in OCR-related research.** Prior works in text generation [1] and recognition [2][3] commonly employ synthetic data to rebalance character categories. Notably, [1][2] train entirely on synthetic data and still achieve state-of-the-art performance when evaluated directly on real handwriting.
>
> **Second, we compare the performance of DiffInk with and without synthetic data augmentation.** As shown in the table below, the decrease in sequence-level DTW is negligible, while the character recognition rates (AR and CR) improve substantially, with gains exceeding 14%. The style score also increases by more than 8%. These results indicate that our data synthesis strategy effectively alleviates the long-tail character distribution in the original dataset, enabling the model to better learn embeddings for rare characters and thereby improving both content recognition and style modeling performance.
>
> ### Effect of Data Augmentation on DiffInk’s Quantitative Performance
>
> | **Training data**        | **AR% ↑** | **CR% ↑** | **Style ↑** | **DTW ↓** |
> |--------------------------|-----------|-----------|-------------|-----------|
> | Original data            | 80.76     | 81.58     | 68.95       | 1.047     |
> | **Original + Augment (Ours)**       | **94.38**     | **94.58**     | **77.38**       | **1.049**     |
>
> More importantly, **DiffInk’s advantage fundamentally stems from its end-to-end text-line modeling paradigm**. As long as synthetic data are provided in the form of full text lines, they positively contribute to model training; some unnatural character transitions function as a form of data augmentation, improving robustness in real-world scenarios.
>
> ---
>
> References:
>
> [1] Pippi V, Quattrini F, Cascianelli S, et al. Zero-Shot Styled Text Image Generation, but Make It Autoregressive. In CVPR 2025.
>
> [2] Li H, Li J, Cao J, et al. Towards Scalable Training for Handwritten Mathematical Expression Recognition[J]. arXiv preprint arXiv:2508.09220, 2025.
>
> [3] Peng, Dezhi, et al. Towards fast, accurate and compact online handwritten Chinese text recognition. In ICDAR 2021.

---

> ### Author Response · Authors · 2025-11-24
> **Response to Reviewer 7DmP [4/4]**
>
> - *Q4: Regarding the reviewer’s concern about the comparison of generation speed.*
>
> The generation speed (Avg. char/s) is computed under the following settings: The batch size is fixed to 1 for all methods , and all methods are required to generate the same set of characters. All experiments are conducted on the same RTX A6000 GPU. The timing starts at the entry point of model inference and excludes data loading and preprocessing time.  The timing window begins at the entry point of model inference and ends when the final character is produced, excluding data loading and preprocessing. The computation process is as follows:
>
> **1). Measure Total Inference Duration**. Record the overall time required for the model to complete all character generation process.
>  **2). Count Generated Characters.** Determine the total number of characters produced during the measurement window.
> **3)Compute Average Speed (Avg. char/s).** Obtain the generation speed by dividing the total character count by the total inference duration.
>
> Importantly, our reported generation speed includes both the DDPM sampling stage and the VAE decoding stage. The speed advantage primarily stems from our text-line–level modeling strategy and the temporal compression mechanism in InkVAE, which allow DiffInk to generate an entire text line within a single inference pass. In contrast, baseline methods generally adopt a per-character generation pipeline, where each forward pass produces only one character. For instance, OLHWG employs a full DDPM sampling schedule to generate each individual character, as this is required for achieving satisfactory visual quality. In contrast, DiffInk adopts a 5-step DDIM sampler, which is sufficient to reach state-of-the-art generation quality in our setting. As a result, OLHWG incurs substantially higher inference cost per character and is considerably slower (0.07 char/s) than DiffInk(58.47 char/s) under identical hardware conditions.
>
> To evaluate the impact of **DDIM sampling steps on generation performance**, we conducted an additional experiment. As shown in the table below, We observe that the performance of DDIM sampling continues to improve as the number of steps increases from 1 to 5. However, when the number of steps exceeds 10, the gains in content and style scores become marginal, while the DTW value increases and the sampling cost grows significantly. Therefore, we adopt 5-step DDIM sampling in this paper as a practical trade-off between generation quality and computational efficiency. We plan to add this part of the analysis in the appendix for completeness.
>
> ### Quantitative Analysis of Different DDIM Sampling Steps.
>
> | **Steps** | **AR% ↑** | **CR% ↑** | **Style ↑** | **DTW ↓** | **Time Cost** |
> |-----------|-----------|-----------|-------------|-----------|---------------|
> | 1         | 36.23     | 36.39     | 35.85       | 1.028     | 0.2x          |
> | 2         | 79.70     | 80.00     | 66.31       | 1.022     | 0.4x          |
> | 3         | 89.93     | 90.19     | 73.51       | 1.026     | 0.6x          |
> | 4         | 93.16     | 93.39     | 76.47       | 1.040     | 0.8x          |
> | **5 (Ours)**     | **94.38** | **94.58** | **77.38**   | **1.052** | **1x**        |
> | 6         | 94.86     | 95.01     | 77.69       | 1.058     | 1.2x          |
> | 10        | 95.35     | 95.52     | 78.32       | 1.067     | 2.0x          |
> | 15        | 95.36     | 95.56     | 79.10       | 1.068     | 3.0x          |
> | 20        | 95.27     | 95.47     | 79.16       | 1.076     | 4.0x          |
>
> ---
> We hope these responses sufficiently address the reviewer’s concerns and are happy to provide further clarification if needed. We sincerely thank the reviewers once again for the thoughtful feedback, which has helped us further improve this work.

---

> ### Comment · Reviewer_7DmP · 2025-11-25
>
> Thank you for the responses, which solve part of my concerns. However, I still have several left questions/comments:
>
> 1. Regarding the method’s novelty, I remain unconvinced that the technical contribution is substantial. Although the architecture, e.g., the content encoder differs from prior work, the paper does not provide compelling evidence that these changes yield clear improvements over existing approaches. In my view, the more significant contribution lies in the application itself: the authors have successfully realized an engaging application. I therefore agree with their claim that this work offers value to the handwritten generation community.
>
> 2. Regarding the baselines in W2.  Thank you for the efforts, but my original intent is to confirm that the method’s performance on full line text generation has been sufficiently validated. I noticed that some strong single character generation methods (e.g., SDT) achieve only mediocre AR / CR performance on full-line text generation, therefore I just suggest evaluating one character generation as a special case of line generation, which is also reasonable.
>
> 3. **Regarding the data augmentation part**.  Prior work (the author list) applies data augmentation only **at the input side**  (*noisy input, clean output*) which can improves the model’s robustness, **but not the reconstruction target itself** (*clean input, noisy output*).  In this paper, one key strength is the ability to model the connection of handwriting, however, concatenating isolated characters as the reconstruction target intuitively breaks that continuity.
>
>
>     The ablation study alone is, in my view, insufficient to certify that the augmentation can be harmless. The augmentation is meant  to address the long tail distribution of the dataset, removing it may simply leave the model under trained. An objective test would be to restrict the vocabulary to those classes that possess sufficient real samples, then train and compare models under equal sized sets of real versus synthetic samples.
>
>
>     However, given the tight rebuttal schedule, dataset indeed has constraints and the performance is relatively satisfying, the current use of augmentation is **understandable**, but I just encourage the authors to revisit this issue carefully in future work.
>
>
> 4. The authors’ statement is otherwise convincing while my only remaining doubt is whether the single character generation baselines can also be accelerated with DDIM？To the best of my knowledge, any model trained with a standard DDPM objective should be compatible with DDIM sampling？

---

> ### Author Response · Authors · 2025-11-29
> **Response to Reviewer 7DmP [1/5]**
>
> *Q1: Novelty of the Proposed Method*
>
> We sincerely appreciate your recognition of the significance of our work, namely that the proposed approach makes a meaningful contribution to the field of handwritten generation. This assessment is also consistent with the feedback from all other reviewers.
>
> From a technical standpoint, **DiffInk is far more than a simple combination of a VAE and a DiT model**. We would like to further clarify the originality of our method from the following three perspectives:
>
> 1. **The First End-to-End Framework for Online Handwritten Text-Line Generation**
>
> Prior to this work, most studies in handwriting generation remained limited to single-character synthesis. In contrast, generating full handwritten text lines is widely regarded as a substantially more challenging problem because the model must simultaneously ensure content accuracy and global style consistency, which places significantly higher demands on long-sequence modeling and generation efficiency. **As noted by you and reviewers p6M8, Kr6c, our paper addresses an important and previously under-explored research objective**: we propose **DiffInk**, the **first** end-to-end framework for handwritten text-line generation, which extends handwriting synthesis from isolated characters to full lines of text. This provides a direct and effective solution to the complex problem of generating long, coherent handwritten sequences. As you also mentioned, such an end-to-end paradigm opens a promising avenue for further advancements in the handwriting generation community.
>
> 2. **InkVAE: A Structured, Task-Aware Latent Space (Non-Standard VAE Architecture)**
>
> In fact, **designing an effective VAE that is suitable for downstream latent diffusion modelling is non-trivial**. As shown in the first row of the Table below, training InkDiT on the latents of a standard reconstruction-only VAE yields only moderate generation quality: AR 74.77\%, CR 75.20\%, and Style score 60.68.
> Although this baseline VAE achieves near-perfect reconstruction (AR/CR $\geq$ 97, Style $\geq$ 99), its latent space is not semantically structured for conditional generation.
>
> To address this challenge, we propose InkVAE, a **novel glyph- and style-aware VAE** designed to efficiently encode long text-line inputs and provide a unified latent representation for text-line generation. By introducing glyph and style regularization losses, we explicitly guide the VAE to learn a latent space that is semantically meaningful and discriminative in terms of content and handwriting style. The feature-space visualizations (Fig. 2 in the paper) clearly demonstrate the effectiveness of this regularization: **content and style are well separated in the latent space**, providing stable and structured conditioning signals for the diffusion generator. **With this properly regularized latent space, InkDiT attains AR 94.38\% ($+$19.61) and CR 94.58\% ($+$19.38) and a Style score of 77.38 ($+$ 16.70)**, as shown in the last row of the table.
>
> ### Ablation Study of InkVAE and Its Impact on InkDiT Generation
>
> |   | **VAE Losses** |  |  | **VAE Reconstruction Performance** |  |  |  | **InkDiT Generation Performance** |  |  |  |
> |-------------|----------------|--|--|------------------------------------|-----------|-----------|-----------|------------------------------------|-----------|-----------|-----------|
> |             | **L_rec+kl** | **L_ocr** | **L_sty** | **AR% ↑** | **CR% ↑** | **Style ↑** | **DTW ↓** | **AR% ↑** | **CR% ↑** | **Style ↑** | **DTW ↓** |
> | Vanilla VAE | ✓ | × | × | 97.59 | 97.61 | 99.97 | 0.014 | 74.77 | 75.20 | 60.68 | 1.062 |
> | Vanilla VAE + Glyph Regularization | ✓ | ✓ | × | 97.60 | 97.63 | 99.97 | 0.015 | 82.09 | 82.41 | 66.07 | 1.052 |
> | Vanilla VAE + Style Regularization | ✓ | × | ✓ | 97.59 | 97.62 | 99.98 | 0.016 | 79.64 | 79.95 | 68.99 | 1.059 |
> | InkVAE (Ours) | ✓ | ✓ | ✓ | 97.65 | 97.67 | 99.97 | 0.016 | **94.38** | **94.58** | **77.38** | **1.049** |
>
> **These results confirm the effectiveness of our task-aware latent regularization. Reviewers p6M8 and Kr6c also noted this, highlighting its applicability to related tasks.**

---

> ### Author Response · Authors · 2025-11-29
> **Response to Reviewer 7DmP [2/5]**
>
> - *Following up on Q1: Novelty of the Proposed Method*
>
> 3. **InkDiT: An Efficient Conditional Latent Diffusion for Handwritten Text-Line Generation**
>
> Building on top of InkVAE, we further design InkDiT, a **novel latent-space diffusion Transformer** that generates high-quality stylized handwritten text lines conditioned on both textual content and writing style. InkDiT incorporates a **lightweight content encoder** that adaptively projects the input text sequence into the latent feature space, ensuring that long text lines are modeled efficiently. The **style conditions are extracted directly from the pretrained InkVAE encoder**, eliminating the need for an additional standalone style encoder and enabling efficient conditional diffusion modeling within a unified latent representation space.
>
> Our ablation studies demonstrate that the lightweight content encoder substantially improves content accuracy in the generated text while introducing minimal computational overhead. As shown in the table below, although this module incurs a slight reduction in generation speed (from 71.94 char/s to 58.47 char/s), it leads to a significant gain in both content accuracy and style consistency: **the average content accuracy increases from 81\% to 94\% (+13\%), and the style score improves from 65.49 to 77.38.**
>
> ### Table: Ablation study of the content encoder in InkDiT.
>
> | **Model**             | **AR% ↑** | **CR% ↑** | **Style ↑** | **DTW ↓** | **Avg. char/s ↑** |
> |-----------------------|-----------|-----------|-------------|-----------|--------------------|
> | w/o ConvNeXtV2        | 81.51     | 81.71     | 65.49       | 1.052     | **71.94**          |
> | **DiffInk (Ours)**    | **94.38** | **94.58** | **77.38**   | **1.049** | 58.47              |
>
> Finally, we would like to emphasize the broader contributions of our work to the community:
>
> (1) **Rigorous Validation:** Extensive ablations and visual analyses validate each component of DiffInk, confirming that our integrated design is both **necessary and effective** for the task.
>
> (2) **Commitment to Open Research:** As appreciated by reviewer Kr6c, our code will be released to support reproducibility and foster further innovation in handwriting synthesis.
>
> (3) **Generalizable Methodology:** Beyond providing a new solution for full-line generation, **our study offers a generalizable perspective on VAE design**. As illustrated in Fig.2 in the main text, structured latent regularization leads to semantically meaningful latent spaces. This idea is also supported by concurrent work such as [1][2][3], which adopts similar regularization strategies in broader image-generation contexts. We believe \**our framework and insights will inspire and enable future research across related generative tasks.**
>
> References:
>
> [1] Palmdiff: When palm-print generation meets controllable diffusion model. In TIP 2025.
>
> [2] Yao J, Yang B, Wang X. Reconstruction vs. generation: Taming optimization dilemma in latent diffusion models. In CVPR 2025.
>
> [3] Guo J, Chen Z, Li W, et al. Compression-aware one-step diffusion model for jpeg artifact removal. In ICCV 2025.

---

> ### Author Response · Authors · 2025-11-29
> **Response to Reviewer 7DmP [3/5]**
>
> - *Q2: On the Baseline Comparison*
>
> Thank you very much for your feedback. All comparison methods in our paper are implemented using their official open-source codebases. Regarding your comment on the SDT[1] method achieving only moderate text accuracy on text-line generation, we would like to clarify an important evaluation difference:
>
> In the original SDT paper, the generated online trajectories are **first rendered into offline images**, and these images are then recognized by an OCR engine. This evaluation protocol is notably more tolerant than those used in methods [2][3][4] as well as ours, since **we evaluate directly on the raw online sequences without rendering**.
>
> To ensure a fair comparison and to demonstrate the superiority of our approach, we rendered both DiffInk and SDT[1] outputs into offline images and evaluated them using PaddleOCR[5]. As shown in the table, SDT’s recognition scores improve under this more robust evaluation pipeline and approach those reported in its original paper. However, they still fall short of our DiffInk, demonstrating the robustness and effectiveness of our method.
>
>
> ### Table: Evaluation of recognition accuracy for offline handwritten text-line images
>
> | **Method**         | **AR% ↑** | **CR% ↑** |
> |--------------------|-----------|-----------|
> | SDT                | 89.21     | 89.50     |
> | **DiffInk (Ours)** | **94.89** | **95.12** |
>
> Additionally, regarding the suggestion to treat single-character generation as a special case of text-line generation, this perspective does not align with the main focus of our study. Text-line generation inherently encompasses richer structural dependencies and long-range consistency, and its recognition performance naturally reflects the per-character accuracy in a more holistic and efficient manner than isolated single-character testing.
>
> References:
>
> [1] Dai G, Zhang Y, Wang Q, et al. Disentangling writer and character styles for handwriting generation. In CVPR 2023.
>
> [2] Tang, Shusen, and Zhouhui Lian. Write Like You: Synthesizing Your Cursive Online Chinese Handwriting via Metric‐based Meta Learning. In Computer Graphics Forum 2021.
>
> [3] Ren, Min-Si, Yan-Ming Zhang, and Yi Chen. Decoupling layout from glyph in online Chinese handwriting generation. In ICLR 2025.
>
> [4] Tang, Shusen, et al. FontRNN: Generating Large‐scale Chinese Fonts via Recurrent Neural Network. In Computer Graphics Forum 2019.
>
> [5] PaddlePaddle Team. PaddleOCR. https://github.com/PaddlePaddle/PaddleOCR, 2025.

---

> ### Author Response · Authors · 2025-11-29
> **Response to Reviewer 7DmP [4/5]**
>
> - *Q3: Regarding the Potential Effects of Synthetic Data*
>
> We analyze the necessity and rationale of incorporating synthetic data in our work, as well as its impact on the overall model performance.
>
> ---
>
> **1. Synthetic Data as an Effective and Common Strategy for Data Augmentation**
>
> Due to the limitations of real-world handwriting data in terms of both scale and diversity, many OCR-related generation and recognition methods leverage synthetic data for augmentation. Synthetic data often provide a more balanced character distribution and richer stylistic variability, effectively compensating for the deficiencies of real datasets. Among existing approaches, some methods [1,2] train directly on large-scale synthetic datasets and achieve state-of-the-art performance when evaluated on real handwriting. In addition, a more common practice adopted in prior work [3,4] is to construct hybrid training sets that combine real and synthetic data, thereby improving model performance and robustness in real-world inference scenarios. Following this paradigm, our DiffInk model is also trained using a mixture of real and synthetic data and is ultimately evaluated on real handwriting.
>
> ---
>
> **2. Analyzing the Impact of Synthetic Data on Model Performance**
>
> By analyzing the real handwritten text-line dataset, we observe a pronounced long-tail distribution across character categories: each writer exhibits substantial variation in character usage and writing frequency. As a result, common characters appear frequently, whereas rare or less commonly used characters occur only a few times. Such a long-tail distribution leads to insufficient learning of content representations for low-frequency characters, ultimately degrading the model’s generation capability.
>
> To mitigate this issue, following prior work [3,4,5], we incorporate synthetic data to compensate for the imbalance in character frequency. Specifically, we construct a mixed training set that combines both real and synthetic text lines. We train the model under different data settings and evaluate all variants on the same real-world test set. As shown in the table, the model trained solely on real data achieves only moderate performance (average content accuracy of 81\% and style score of 68.95). In contrast, **augmenting the training set with synthetic data substantially improves performance: content accuracy increases to 94\% (+13\%), and the style score increases to 77.38 (+8.43).** These results indicate that synthetic data effectively alleviates the character long-tail issue, enabling the model to better learn character-level representations. Moreover, exposure to more diverse character combinations enhances the model’s robustness in style modeling.
>
> ### Table: Performance of **DiffInk (Ours)** Under Different Training Data Configurations
>
> | **Training data**      | **Real-Data Fine-Tuning** | **AR% ↑** | **CR% ↑** | **Style ↑** | **DTW ↓** |
> |------------------------|---------------------------|-----------|-----------|-------------|-----------|
> | Real Data Only         | No                        | 80.76     | 81.58     | 68.95       | 1.047     |
> | Real + Augment Data   | No                        | 94.38     | 94.58     | 77.38       | 1.049     |
> | Real + Augment Data    | Yes                       | 94.56     | 94.71     | 78.11       | 1.041     |
>
> ---
>
> **3. Fine-tuning the DiffInk (Ours) on real data**
>
> We also understand the reviewer’s concern regarding the potential negative impact of synthetic data on temporal continuity modeling. **To address this issue, we designed a simple yet effective mitigation strategy.** During training, we continue to use a mixture of real and synthetic data, ensuring that characters in the long-tail distribution are sufficiently learned. **After the model converges, we perform a brief fine-tuning stage using only real data (about 1,000 steps)**, allowing the model to quickly adapt to the real-data distribution.
>
> We then evaluate the fine-tuned model on the real test set. As shown in the table, **the style score further increases from 77.38 to 78.11 (+0.73)**, indicating that **the potential adverse effects introduced by synthetic data can be effectively mitigated through lightweight fine-tuning on real handwriting.** This strategy is also consistent with prior studies[4], which similarly adopt real-data fine-tuning to mitigate the domain gap between synthetic and real samples.
>
> ---
>
> In summary, the use of **synthetic data is necessary** because it substantially **alleviates the long-tail sparsity inherent in real handwriting datasets.**
> Although synthetic samples inevitably differ from real data in distribution, their use introduces only **limited adverse effects** when combined with real data during training. Moreover, a brief fine-tuning stage on real data can effectively mitigate potential effects introduced by synthetic data, enabling the model to better align with the real data distribution during inference.

---

> ### Author Response · Authors · 2025-11-29
> **Response to Reviewer 7DmP [5/5]**
>
> We sincerely thank the reviewer for the insightful comments on this part. We will incorporate the corresponding analysis in the revised manuscript to make our study and its discussions more complete and comprehensive.
>
> References related to question 3:
>
> [1] Pippi V, Quattrini F, Cascianelli S, et al. Zero-Shot Styled Text Image Generation, but Make It Autoregressive. In CVPR 2025.
>
> [2] Li H, Li J, Cao J, et al. Towards Scalable Training for Handwritten Mathematical Expression Recognition. arXiv preprint arXiv:2508.09220, 2025.
>
> [3] Chang J H R, Bresler M, Chherawala Y, et al. Data incubation—synthesizing missing data for handwriting recognition. In ICASSP 2022.
>
> [4] Kang L, Rusinol M, Fornés A, et al. Unsupervised writer adaptation for synthetic-to-real handwritten word recognition. In CVPR 2020.
>
> [5] Zhang Y, Shi Y, Zhang P, et al. MegaHan97K: A large-scale dataset for mega-category Chinese character recognition with over 97K categories. In Pattern Recognition 2025.
>
> ---
>
>
> - *Q4: Replacing DDPM with DDIM for Faster Inference*
>
> We accelerate OLHWG[1] by replacing its original DDPM[2] scheduler with a standard DDIM[3] sampler, and then compare its inference speed and generation quality against our method.
>
> Specifically, our method exhibits substantially better performance under the same DDIM acceleration setting. With an equal number of DDIM steps (5), **DiffInk (Ours) achieves much higher generation throughput (58.47 vs. 14.00 characters/s) and significantly better content accuracy and style consistency (OLHWG: ~82\% content accuracy, ~17.61 style score; Ours: ~94\% content accuracy, ~77.38 style score).** These results highlight the robustness and effectiveness of our design.
>
> Benefiting from line-level modeling and the temporal compression provided by InkVAE, **DiffInk (Ours) is able to generate an entire handwritten text line within only 5 DDIM steps.** This enables substantial speedup while maintaining high-quality and structurally consistent generation results. We thank the reviewer for highlighting this point and will include the corresponding analysis in the appendix. This further demonstrates the effectiveness of our method in both generation quality and inference efficiency.
>
> ---
>
> ### Improving Diffusion Sampling Efficiency with DDIM[3]
>
> | **Method**         | **Sampler** | **Steps** | **AR% ↑** | **CR% ↑** | **Style ↑** | **DTW ↓** | **Avg. char/s ↑** |
> |--------------------|-------------|-----------|------------|------------|-------------|-----------|--------------------|
> | OLHWG[1]              | DDPM        | 1000      | 91.48      | 91.71      | 44.74       | 1.326     | 0.07               |
> | OLHWG[1]              | DDIM        | 100       | 87.67      | 87.91      | 25.46       | 1.319     | 0.69               |
> | OLHWG[1]              | DDIM        | 50        | 87.09      | 87.33      | 26.45       | 1.320     | 1.39               |
> | OLHWG[1]              | DDIM        | 20        | 88.59      | 88.75      | 24.47       | 1.325     | 3.45               |
> | OLHWG[1]              | DDIM        | 5         | 82.69      | 82.93      | 17.61       | 1.408     | 14.00              |
> | **DiffInk (Ours)** | **DDIM**    | **5**     | **94.38**  | **94.58**  | **77.38**   | **1.049** | **58.47**          |
>
> ---
>
> We once again thank the reviewer for the thoughtful and insightful comments, which have substantially improved the quality of our work. We hope the additional results and explanations address your concerns. We are glad to clarify any remaining issues if necessary.
>
> ---
>
> References:
>
> [1] Ren, Min-Si, Yan-Ming Zhang, and Yi Chen. Decoupling layout from glyph in online Chinese handwriting generation. In ICLR 2025.
>
> [2] Ho J, Jain A, Abbeel P. Denoising diffusion probabilistic models. In NeurIPS 2020.
>
> [3] Song J, Meng C, Ermon S. Denoising diffusion implicit models. In ICLR 2021.

---

### Official Review · Reviewer_U8jE · 2025-10-29

**Soundness:** 2
**Presentation:** 3
**Contribution:** 2
**Rating:** 4
**Confidence:** 4

**Summary:**

The paper presents a well-executed study with significant contributions to the task of end-to-end handwritten text-line generation, demonstrating both theoretical and practical advancements. While the empirical results are strong and the methodology is clearly justified, there are certain limitations in the experimental setup, data synthesis approach, and the exploration of real-world implications. Addressing these concerns would strengthen the work and provide a more comprehensive understanding of the method's impact and limitations.

If the authors can address my doubts or point out areas I may have overlooked, I would be willing to reconsider my evaluation and potentially improve my score.

**Strengths:**

1. **Significant Empirical Advancement in a Challenging Task**
   The paper demonstrates a notable and convincing empirical improvement on the relatively underexplored and complex task of end-to-end online handwritten text-line generation. The proposed DiffInk framework achieves superior performance across both quantitative metrics (Content AR/CR, Style Accuracy, DTW) and qualitative visual comparisons, thereby setting a new state-of-the-art benchmark in this domain.

2. **Clear, Well-Structured, and Justified Methodology**
   The overall framework is presented with exceptional clarity. The design of the two core components, InkVAE and InkDiT, is both logically sound and well-motivated. The introduction of task-specific regularization losses (OCR, style) to structure the latent space offers a direct and effective solution to a clearly identified problem, making the method both easy to follow and reproducible.

3. **Comprehensive and Convincing Ablation Studies**
   The paper is notably supported by thorough ablation studies that systematically validate the contribution of each key component (e.g., InkVAE regularization losses, InkDiT's content encoder). These experiments provide strong evidence that the observed performance gains are directly attributable to the proposed architectural choices.

4. **Noteworthy Practical Efficiency**
   In addition to its quality, the method achieves a dramatic speedup in generation (e.g., >800x faster than OLHWG), making it not only more effective but also significantly more practical for real-world applications.

**Weaknesses:**

1. **Incremental Nature of the Core Technical Contribution**
   While the empirical results are robust, the core technical innovation can be characterized more as a skillful integration and refinement of existing components (VAE, DiT) rather than a groundbreaking algorithmic advancement.

2. **Fundamental Limitation in Training Data Synthesis**
   The reliance on a data augmentation strategy that stitches isolated characters into text lines presents a significant methodological limitation. This approach inherently fails to capture the cursive connections and natural co-articulation found in real handwritten text, which likely imposes a ceiling on the model’s ability to generate fully authentic, fluid text lines and raises questions about the underlying sources of its performance gains.

3. **Potential Bias in the Experimental Setup**
   The decision to compare the method against character-level models extended by a single, shared layout prediction module (from OLHWG) is a practical but potentially unbalanced choice for the baselines. This setup may unfairly attribute the weaknesses of this specific layout module to the baseline generation methods, thus exaggerating DiffInk's advantages in terms of coherence and layout.

4. **Limited Exploration of Practical Impact and Downstream Utility**
   The paper would benefit from a more comprehensive discussion of its real-world implications. While the OCR data augmentation experiment is a step in the right direction, it uses a relatively weak baseline model, which diminishes the argument that the synthetic data is of sufficiently high quality to improve modern, high-performance systems.

**Questions:**

Please refer to the weakness.

---

> ### Author Response · Authors · 2025-11-24
> **Response to Reviewer U8jE [1/3]**
>
> We thank the reviewer for the thoughtful comments and the valuable feedback. We are pleased that the reviewer recognizes our work as making an important contribution to end-to-end handwritten text-line generation, achieving compelling results on this challenging task, and providing clear module motivations, well-structured methodological descriptions, and an approach that is easy to reproduce and understand.
>
> We address the reviewer’s concerns and questions as follows.
>
> ---
>
> - *Q1: Regarding the reviewer’s concern about the incremental nature of the technical contributions.*
>
> DiffInk is not a simple concatenation of a VAE and a DiT model; rather, it represents a task- related design that provides an effective solution tailored for online handwritten text-line generation. The novelty of our work can be summarized along two dimensions: research motivation and technical contributions.
>
> **1) Research Motivation**:
>
> Prior to this work, most studies in handwriting generation focused on isolated character synthesis, while full text-line generation was considered highly challenging. Our work introduces a paradigm shift from character-level modeling to **end-to-end text-line modeling**. To the best of our knowledge, DiffInk is the **first** unified framework specifically designed for efficient, end-to-end handwritten text-line generation, offering a more direct and innovative solution for producing complex long-sequence handwriting. As the reviewer Kr6c mentioned, we believe this novel end-to-end generation approach will benefit the overall development of the community, and we will release the code to facilitate future research and accessibility.
>
> **2) Technical Contributions:**
>
> **Designing an efficient VAE for downstream conditional synthesis of handwritten text lines is a non-trivial task**, as shown in our ablation studies where simply combining a vanilla VAE with a conditional diffusion model results in only moderate character accuracy and style consistency. To address this issues, **we introduce novel glyph-aware and style-aware regularization strategies within InkVAE**, substantially improving the stability and performance of the latent diffusion model used in downstream generation. Building upon InkVAE, we design InkDiT, a latent-space diffusion Transformer conditioned on textual content and handwriting style to produce high-fidelity stylized text lines. Extensive comparisons and ablation studies verify the soundness and effectiveness of the DiffInk design. **As the reviewer Kr6c, p6M8** also highlighted, the task-specific regularization strategies proposed in InkVAE offer broadly applicable insights that are beneficial for related handwriting generation tasks.
>
> ### Ablation on VAE Losses and InkDiT Generation Performance
>
> | **VAE Losses** |        |        | **InkDiT Generation Performance** |        |        |        |
> |----------------|--------|--------|-----------------------------------|--------|--------|--------|
> | **L_rec+kl**   | **L_ocr** | **L_sty** | **AR% ↑** | **CR% ↑** | **Style ↑** | **DTW ↓** |
> | ✓              | ×      | ×      | 74.77     | 75.20     | 60.68     | 1.062     |
> | ✓              | ✓      | ×      | 82.09     | 82.41     | 66.07     | 1.052     |
> | ✓              | ×      | ✓      | 79.64     | 79.95     | 68.99     | 1.059     |
> | ✓              | ✓      | ✓      | **94.38** | **94.58** | **77.38** | **1.049** |
>
> Overall, existing approaches [1][2] often rely on multi-stage pipelines to approximate text-line generation. In contrast, **DiffInk achieves end-to-end full-line generation with a concise and efficient design, offering a new perspective and practical methodology for this challenging problem.**
>
> [1] Ren M S, Zhang Y M, Chen Y. Decoupling layout from glyph in online Chinese handwriting generation. In ICLR 2025.
>
> [2] Template-Guided Cascaded Diﬀusion for Stylized Handwritten Chinese Text-Line Generation. In ICDAR 2025.

---

> ### Author Response · Authors · 2025-11-24
> **Response to Reviewer U8jE [2/3]**
>
> - *Q2: Regarding the reviewer’s concern about the use of synthetic data.*
>
> **First, we would like to emphasize that the use of synthetic data as a data augmentation strategy is widely adopted in OCR-related research**. Prior works in text generation [1] and recognition [2][3] commonly employ synthetic data to rebalance character categories. Notably, [1][2] train entirely on synthetic data and still achieve state-of-the-art performance when evaluated directly on real handwriting.
>
> **Second, we compare the performance of DiffInk with and without synthetic data augmentation.** As shown in the table below, the decrease in sequence-level DTW is negligible, while the character recognition rates (AR and CR) improve substantially, with gains exceeding 14%. The style score also increases by more than 8%. These results indicate that our data synthesis strategy effectively alleviates the long-tail character distribution in the original dataset, enabling the model to better learn embeddings for rare characters and thereby improving both content recognition and style modeling performance.
>
> ### Effect of Data Augmentation on DiffInk’s Quantitative Performance
>
> | **Training data**        | **AR% ↑** | **CR% ↑** | **Style ↑** | **DTW ↓** |
> |--------------------------|-----------|-----------|-------------|-----------|
> | Original data            | 80.76     | 81.58     | 68.95       | **1.047**     |
> | **Original + Augment (Ours)**       | **94.38**     | **94.58**     | **77.38**       | 1.049     |
>
> More importantly, **DiffInk’s advantage fundamentally stems from its end-to-end text-line modeling paradigm**. As long as synthetic data are provided in the form of full text lines, they positively contribute to model training; some unnatural character transitions function as a form of data augmentation, improving robustness in real-world scenarios.
>
> ---
>
> - *Q3: Regarding the evaluation setup for baseline models.*
>
> In response to the reviewer’s concern that inaccurate bounding-box predictions may exaggerate the advantage of DiffInk, we conducted an additional experiment where character-level baselines are stitched using the ground-truth bounding boxes from real text lines. As shown in the table below, the content scores change only marginally because the OCR model is largely insensitive to character positions. The writer-identification model, which is sensitive to both local and global spatial layout, shows performance improvements when given ground-truth positions. Importantly, DiffInk still maintains a clear advantage under this setting.
>
> ### Quantitative comparison. All character-level generation methods obtain their text-line results through post-processing using ground-truth character bounding boxes.
>
> | **Method**   | **Venue**  | **Output**      | **AR% ↑** | **CR% ↑** | **Style ↑** | **DTW ↓** |
> |--------------|------------|-----------------|-----------|-----------|-------------|-----------|
> | Drawing      | TPAMI'17   | Char-level      | 76.91±0.56     | 77.62±0.75     | 28.32±2.57       | 1.521±0.061     |
> | DeepImitator | PR'20      | Char-level      | 79.10±0.89     | 79.35±0.37     | 29.47±1.85       | 1.503±0.058     |
> | WLU          | CGF'21     | Char-level      | 80.51±0.66     | 80.97±0.74     | 36.81±1.1       | 1.489±0.054     |
> | SDT          | CVPR'23    | Char-level      | 83.21±0.68     | 83.50±0.5     | 52.83±2.32       | 1.214±0.056     |
> | OLHWG        | ICLR'25    | Char+layout     | 91.66±0.18     | 91.92±0.21     | 46.21±1.47       | 1.285±0.041     |
> | **DiffInk**  | **This work** | **Line-level** | **94.38** | **94.58** | **77.38**   | **1.049** |
>
> Fundamentally, this phenomenon reflects the distributional mismatch between isolated-character data and real text-line data. DiffInk’s superior performance arises from its end-to-end text-line modeling paradigm—a conclusion supported by both our visualizations and quantitative metrics.
>
> We also emphasize that ground-truth text-line bounding boxes are not available during inference, making this evaluation protocol unrealistic for real-world deployment. The testing strategy used in the paper reflects the actual scenario and provides a fair comparison at the text-line level.
>
> We appreciate the reviewer’s attention to evaluation fairness, and we would welcome further suggestions if you have additional insights on how to improve the assessment protocol.
>
> ---
>
> References
>
> [1] Pippi V, Quattrini F, Cascianelli S, et al. Zero-Shot Styled Text Image Generation, but Make It Autoregressive. In CVPR 2025.
>
> [2] Li H, Li J, Cao J, et al. Towards Scalable Training for Handwritten Mathematical Expression Recognition[J]. arXiv preprint
> arXiv:2508.09220, 2025.
>
> [3] Peng, Dezhi, et al. Towards fast, accurate and compact online handwritten Chinese text recognition. In ICDAR 2021.

---

> ### Author Response · Authors · 2025-11-24
> **Response to Reviewer U8jE [3/3]**
>
> - *Q4: Regarding the limited exploration of downstream tasks.*
>
> Although the main focus of this work is stylized handwritten text-line synthesis, we place strong emphasis on practical applicability. Due to the lack of publicly available text-line–level handwriting recognition methods, we reproduced the approach from [1] as a stronger OCR baseline. We trained this model on both the original dataset and an augmented version that includes DiffInk-generated samples. The results show that incorporating our synthetic data consistently improves OCR performance, indicating that DiffInk produces high-quality handwriting samples that are effective for data augmentation and can enhance downstream recognition tasks.
>
> | **OCR Model**             | **Training Data**              | **CR ↑** | **AR ↑** |
> |----------------------------|--------------------------------|:--------:|:--------:|
> | **CNN + Transformer**      | Real data only (Baseline)      | 77.2     | 77.7     |
> |                            | + **DiffInk Synth (Ours)**     | **85.7** | **86.4** |
> | **GLRNet (New added)** | Real data only (Baseline) | 86.2     | 86.6     |
> |                            | + **DiffInk Synth (Ours)**     | **90.9** | **91.1** |
>
> In future work, we plan to further investigate how to select and enhance synthetic text-line data, with the goal of better understanding and maximizing the practical benefits of generation-based augmentation in real-world applications. We also sincerely welcome the reviewer’s recommendations of open-source text-line OCR methods for our further exploration and study.
>
> ---
>
> We hope these responses sufficiently address the reviewer’s concerns and are happy to provide further clarification if needed. We sincerely thank the reviewers once again for the thoughtful feedback, which has helped us further improve this work.
>
> ---
>
> References:
>
> [1] Peng, Dezhi, et al. Towards fast, accurate and compact online handwritten Chinese text recognition. In ICDAR 2021.

---

> > ### Comment · Reviewer_U8jE · 2025-11-24
> >
> > Thank you to the author for the response and additional experiments. The response has addressed most of my concerns, but I still have the following suggestions:
> >
> > 1. When validating the proposed method’s performance on downstream tasks (i.e., textline recognition tasks), is the selected baseline model appropriate? To my best knowledge, on ICDAR2013-Online, most models (including segmentation-based frameworks [1], fully convolutional frameworks [2][3], and LSTM frameworks [4]) can achieve performance around AR: 85; CR: 85 without using a language model. The baseline selected by the authors is slightly below the current standard. Although the current experiments already demonstrate the benefits of the synthesized data for downstream recognition tasks, I am more curious about the upper limit of the current method’s benefits for state-of-the-art (SOTA) models, as using only rule-based methods for data synthesis has already achieved SOTA performance [2][3].
> >
> > 2. The author’s goal is to better understand and maximize the practical benefits of generation-based augmentation in real-world applications, which I highly appreciate. However, I would like to see more diverse discussions beyond this, as the text line generation task itself is also a valuable application, such as in handwriting verification and identity authentication, as well as personalized font generation.
> >
> > Based on the author’s response and supplementary content, I am willing to adopt a more optimistic attitude toward this paper, and therefore, I will raise my score.
> >
> > References:
> >
> > [1] Chen Y, Zhang H, Liu C L. Improved learning for online handwritten Chinese text recognition with convolutional prototype network[C]//International Conference on Document Analysis and Recognition. Cham: Springer Nature Switzerland, 2023: 38-53.
> >
> > [2] Peng D, Xie C, Li H, et al. Towards fast, accurate and compact online handwritten Chinese text recognition[C]//International Conference on Document Analysis and Recognition. Cham: Springer International Publishing, 2021: 157-171.
> >
> > [3] Peng D, Jin L, Ma W, et al. Recognition of handwritten Chinese text by segmentation: a segment-annotation-free approach[J]. IEEE Transactions on Multimedia, 2022, 25: 2368-2381.
> >
> > [4] Xie Z, Sun Z, Jin L, et al. Learning spatial-semantic context with fully convolutional recurrent network for online handwritten chinese text recognition[J]. IEEE transactions on pattern analysis and machine intelligence, 2017, 40(8): 1903-1917.

---

> > ### Author Response · Authors · 2025-12-03
> > **Response to Reviewer U8jE**
> >
> > We sincerely appreciate the reviewer’s recognition of our rebuttal efforts, which improved the score from 4 to 6. Following the reviewer’s suggestions, we conducted the following experiments and improvements.
> >
> > ---
> >
> > - *Suggestion 1: Baseline Selection for OCR Recognition Models*
> >
> > The OCR methods referenced by the reviewers were trained on substantially larger datasets, typically covering the entire CASIA-OLHWDB 2.0–2.2 training set (over 800 writers). As explicitly stated in our paper, to maintain consistency with the training setup of DiffInk, we used **only the subset of approximately 400 writers for training the OCR (CNN+Transformer) model.** This reduced data scale also contributes to the observed performance gap relative to previous works.
> >
> > To quantify the impact of training data scale, we retrained the OCR model using the full CASIA-OLHWDB 2.0–2.2 dataset. The baseline accuracy improved to approximately 84\%. When further incorporating augmented samples generated by DiffInk, recognition accuracy increased from around 84\% to nearly 90\%, demonstrating the practical effectiveness of DiffInk-generated data in enhancing OCR performance.
> >
> > **Table: Recognition performance on ICDAR13 with and without our synthesized data**
> >
> > | **OCR Model**       | **Training Data**                         | **CR ↑** | **AR ↑** |
> > |---------------------|-------------------------------------------|----------|----------|
> > | **CNN+Transformer** | Real data only (subset)                   | 77.2     | 77.7     |
> > |                     | Real data only (full set)                 | 83.9     | 84.2     |
> > |                     | Real data only (full set) + **DiffInk Synth (Ours)** | **89.6** | **90.0** |
> >
> > Due to computational constraints, DiffInk is currently trained on only a subset of the available data, limiting the diversity of augmentation it can provide. In future work, we plan to train DiffInk on all publicly available datasets to yield more broadly distributed synthetic samples and further improve the robustness of OCR models. Moreover, since most state-of-the-art text-line OCR systems are not open-sourced, we also plan to reproduce several recent SoTA methods (including those cited by the reviewer) to more comprehensively assess the upper bound of performance gains that generative models like DiffInk can bring to OCR recognition.
> >
> > ---
> >
> > - *Suggestion 2: Potential Applications Beyond Data Augmentation*
> >
> > We sincerely thank the reviewer for **recognizing the value of our work and for the insightful comments.** We fully agree that generation-based augmentation is only one aspect of the broader impact of text-line generation. Indeed, handwritten text-line synthesis itself has important applications beyond augmentation, such as handwriting verification, identity authentication, writer-specific style modeling, and personalized handwriting generation. Although, due to time constraints, we were unable to conduct experiments for these specific application scenarios, we believe that our proposed framework has the potential to bring new vitality to the handwriting research community.
> >
> >
> > ---
> >
> >
> > Once again, we sincerely appreciate your recognition of our research motivation and the valuable suggestions provided. In the revised manuscript, we plan to expand this section with a more detailed discussion to make our study more comprehensive and thorough.

---

### Official Review · Reviewer_Kr6c · 2025-10-30

**Soundness:** 3
**Presentation:** 4
**Contribution:** 3
**Rating:** 8
**Confidence:** 4

**Summary:**

This paper proposes a latent diffusion Transformer for online handwriting generation. Authors cite speed and the ability of conditioning on the  full line as the main problems addressed.

Two main components are:
* InkVAE: VAE trained with additional losses for recognition (forcing generated samples to be recognizable) and writer-identity-style classification (enforcing style-space disentanglement).
* InkDiT: diffusion Transformer (DiT) in the latent space created by InkVAE, conditioned on the embedding of a target text and style reference passed through InkVAE encoder.

Experiments are done on CASIA-OLHWDB (Chinese) dataset, with evaluation in terms of accuracy, style consistency, and generation speed.

**Strengths:**

* InkVAE design is interesting, novel, and well ablated (ex. Table 2, Figure 2 ablate the auxiliary losses for InkVAE construction). The approach seems generalizable (ex could be similarly applied to cursive handwriting, sketch generation, etc.)

* Quantitative results are quite strong (most impressively in the generation speed, but also on recognition accuracy and style classification). While two-step approach (first generate bounding boxes, then fill the letters) could also allow for high generation speed (as individual characters could be generated in parallel, at least in the case of Chinese), the proposed one-step approach with InkDiT would be superior in cases like cursive and sketches.

* Additional results & ablations (generalizability to English handwriting,use as data augmentation tool for OCR, detailed comparison to two-stage approach) further strengthen the work (in my opinion, they should be more prominently highlighted in the main body - before reading them in the appendix, I assumed the model would generalize poorly beyond Chinese handwriting.

* Assuming the code will be released beside the paper, I believe this will be a significant contribution on which the community can further build.

* The writing is generally very clear, and the set of ablations is comprehensive.

**Weaknesses:**

Limited evaluation: Main evaluation is performed on a single Chinese dataset. Paper could be significantly strengthened by performing evaluation on other domain, such as other languages (BRUSH, IAM), documents (IAMOnDB), sketches (QuickDraw), math (MathWriting), etc.

Style Conditioning: The model is conditioned on a single style vector $x_{ref}$. It is unclear how it would handle more complex or ambiguous style inputs, such as multiple, varied reference samples from the same writer, or a single reference that contains two different styles. Paper could be strengthened by the evaluation of style mixing, etc.

Text conditioning: The content encoder relies on a learnable codebook for a fixed set of characters. This is a standard approach but presents a hard limitation for out-of-vocabulary (OOV) characters, which is a significant issue for languages like Chinese (esp given ~2.6k characters while the alphabet could expand to 30k). Paper could be strengthened by the discussion around this topic, or by evaluation of image-based conditioning for text.

**Questions:**

- Model implicitly learns layout from the style reference. Have you considered a more explicit form of control, such as conditioning the InkDiT on a target baseline curve, bounding boxes, or other spatial parameters?

- The paper claims 800x speedup over the baseline. Does 58.47 char/s figure for DiffInk include the conditioning, the 5-step DDIM sampling and the final VAE decoding step?

- Can the model currently handle OOV characters in any way?

---

> ### Author Response · Authors · 2025-11-24
> **Response to Reviewer Kr6c [1/3]**
>
> We thank the reviewer for the thoughtful comments and positive feedback. Your recognition of the innovations in InkVAE, the potential generality of its regularization strategies, the advantages of DiffInk in both generation efficiency and recognition accuracy, and the completeness of our ablation studies is highly valuable and has helped us further refine the work. As stated in the abstract, we also plan to release the code to facilitate reproducibility and future research within the community.
>
> We address the reviewer’s concerns and questions as follows.
>
> ---
> - *Q1: On the possibility of incorporating explicit layout control*
>
> Thank you for the insightful suggestion. Introducing more explicit layout conditions into the modeling process is indeed a promising direction.
>
> In contrast to explicitly modeling character bounding boxes as layout conditions, a more appropriate and flexible approach is to incorporate lightweight supervisory layout signals. Such as character centroids, which can provide structural guidance without imposing hard geometric constraints. Similar ideas have been validated in text spotting literature [1][2], where replacing bounding boxes with point-level supervision leads to better localization and recognition performance.
> We plan to explore such lightweight and structurally meaningful layout cues in future work, following the ideas of ControlNet [3] and Omni-Control [4]. Based on the simple and effective end-to-end framework of DiffInk, incorporating explicit layout injection modules is a promising future direction.
>
> Reference:
>
> [1] Peng D, Wang X, Liu Y, et al. Spts: single-point text spotting. In ACM MM 2022.
>
> [2] Liu Y, Zhang J, Peng D, et al. Spts v2: single-point scene text spotting. In TPAMI 2023.
>
> [3] Zhang L, Rao A, Agrawala M. Adding conditional control to text-to-image diffusion models. ICCV 2023.
>
> [4] Tan Z, Liu S, Yang X, et al. Ominicontrol: Minimal and universal control for diffusion transformer. In ICCV 2025.

---

> ### Author Response · Authors · 2025-11-24
> **Response to Reviewer Kr6c [2/3]**
>
> - *Q2: Regarding the question on the comparison of generation speed.*
>
> The generation speed (Avg. char/s) is computed under the following settings: The batch size is fixed to 1 for all methods , and all methods are required to generate the same set of characters. All experiments are conducted on the same RTX A6000 GPU. The timing starts at the entry point of model inference and excludes data loading and preprocessing time.  The timing window begins at the entry point of model inference and ends when the final character is produced, excluding data loading and preprocessing. The computation process is as follows:
>
> **1. Measure Total Inference Duration.** Record the overall time required for the model to complete all character generation process.
>
> **2. Count Generated Characters.** Determine the total number of characters produced during the measurement window.
>
> **3. Compute Average Speed (Avg. char/s).** Obtain the generation speed by dividing the total character count by the total inference duration.
>
> Importantly, our reported generation speed **includes both the DDPM sampling stage and the VAE decoding stage.** The speed advantage primarily stems from our text-line–level modeling strategy and the temporal compression mechanism in InkVAE, which allow DiffInk to generate an entire text line within a single inference pass. In contrast, baseline methods generally adopt a per-character generation pipeline, where each forward pass produces only one character. For instance, OLHWG employs a full DDPM sampling schedule to generate each individual character, as this is required for achieving satisfactory visual quality. In contrast, DiffInk adopts a 5-step DDIM sampler, which is sufficient to reach state-of-the-art generation quality in our setting. As a result, OLHWG incurs substantially higher inference cost per character and is considerably slower (0.07 char/s) than DiffInk (58.47 char/s) under identical hardware conditions.
>
> To evaluate the impact of DDIM sampling steps on generation performance, we conducted an additional experiment. As shown in the table below, We observe that the performance of DDIM sampling continues to improve as the number of steps increases from 1 to 5. However, when the number of steps exceeds 10, the gains in content and style scores become marginal, while the DTW value increases and the sampling cost grows significantly. Therefore, we adopt 5-step DDIM sampling in this paper as a practical trade-off between generation quality and computational efficiency. We plan to add this part of the analysis in the appendix for completeness.
>
> ### Quantitative Analysis of Different DDIM Sampling Steps.
>
> | **Steps** | **AR% ↑** | **CR% ↑** | **Style ↑** | **DTW ↓** | **Time Cost** |
> |-----------|-----------|-----------|-------------|-----------|---------------|
> | 1         | 36.23     | 36.39     | 35.85       | 1.028     | 0.2x          |
> | 2         | 79.70     | 80.00     | 66.31       | 1.022     | 0.4x          |
> | 3         | 89.93     | 90.19     | 73.51       | 1.026     | 0.6x          |
> | 4         | 93.16     | 93.39     | 76.47       | 1.040     | 0.8x          |
> | **5 (Ours)**     | **94.38** | **94.58** | **77.38**   | **1.052** | **1x**        |
> | 6         | 94.86     | 95.01     | 77.69       | 1.058     | 1.2x          |
> | 10        | 95.35     | 95.52     | 78.32       | 1.067     | 2.0x          |
> | 15        | 95.36     | 95.56     | 79.10       | 1.068     | 3.0x          |
> | 20        | 95.27     | 95.47     | 79.16       | 1.076     | 4.0x          |
>
> ---
>
> - *W1: limited evaluation*
>
> Thank you for your forward-looking comments. As the first attempt to perform end-to-end generation of complex handwritten text lines, DiffInk demonstrates that combining textual content with style information enables the effective synthesis of long handwriting sequences with coherent structure and consistent style. The current work focuses primarily on stylized generation of unstructured “text lines.” Expanding the framework to support a broader range of handwritten elements—such as structured mathematical expressions, tables, and other document-level components—**will be a key direction for our future research.** We plan to adopt a unified text-encoding strategy to represent diverse forms of textual input and enable the model to generate the corresponding handwritten expressions accordingly. **We also believe that, once accepted and open-sourced, our work will further support research in this area and benefit the community.**

---

> ### Author Response · Authors · 2025-11-24
> **Response to Reviewer Kr6c [3/3]**
>
> - *W2: Can DiffInk Support More Complex Forms of Style Conditioning?*
>
> DiffInk uses a continuous handwritten segment as the style reference to produce a text line with consistent stylistic characteristics, following prior text-line generation approaches [1][2]. However, **DiffInk can support more complex forms of style input**, we conducted an exploration at the feature level. Specifically, we encode reference trajectories from two different writers using the VAE to obtain two latent style representations. These representations are then combined through weighted averaging to form a new mixed-style features, which is used as the contextual condition for generating the subsequent text line. The results are included in Figure 11 on page 19 of the appendix in the revised PDF, demonstrate that DiffInk can successfully support style mixing and generate stylistic variations beyond the original training distribution.
>
>
> ---
>
> - *W3&Q3 — Image-based text conditioning*
>
> Thank you for raising this valuable point. Using image-based encodings to provide structural information for characters has indeed been well explored in character stylization research. However, extending this idea to long and complex text-line generation remains largely underexplored. The main challenge lies in the fact that text lines involve not only character-level structural cues but also positional and size variations—information that is difficult to obtain from existing character templates.
>
> To investigate the feasibility of image-based text conditioning for OOV scenarios in long text lines, we conducted a preliminary exploration. We first prepared a glyph library (Heiti.ttf), covering more than 10k Chinese and English characters, and rendered each required character as an image. We then evaluated two ways of extracting content features from these images:
>
> **1) Full-line template encoding:** Render the entire text line as a single template image and extract its representation using an image encoder. **2) Per-character encoding with spatial concatenation:**  Render each character individually, encode each using an image encoder, concatenate the features in character order along the spatial dimension, and then apply positional encoding to form a text-line representation. All character images were rendered at 128×128 resolution, and the encoder used was a ResNet-18. We retrained DiffInk under both image-based conditioning schemes and evaluated performance accordingly.
>
> ### Quantitative results under different textual conditioning strategies
>
> | **Diffusion’s textual condition**        | **AR% ↑** | **CR% ↑** | **Style ↑** | **DTW ↓** | **Avg. char/s ↑** |
> |------------------------------|-----------|-----------|-------------|-----------|-------------------|
> | Full Image + ResNet-18       | 70.21     | 70.45     | 45.81       | 1.431     | 53.24             |
> | Single Image + ResNet-18     | 86.79     | 86.92     | 65.81       | 1.231     | 50.10             |
> | **Embedding + ConvNeXt-V2 (Ours)**      | **94.38**     | **94.58**     | **77.38**       | **1.049**     | **58.47**             |
>
> From the quantitative results, both image-based approaches underperform the embedding-based method, particularly the full-line template encoding, which increases computational cost while yielding lower accuracy. Although image-based text conditioning is generally more flexible in handling diverse text inputs, **it is also sensitive to template font selection, encoder design, and the way text conditions are aligned with the latent diffusion process**, requiring careful engineering to be effective in text-line generation.
>
> DiffInk directly incorporates textual embeddings into the VAE latent space as conditioning signals. **This approach has been shown to be more effective than designing separate conditional control modules**, as demonstrated in recent studies [3][4]. **It enables content semantics and structural features to be jointly modeled within a unified representation space.** Designing a more general text-conditioning mechanism (e.g., image-based inputs or universal text-encoder inputs) will be an important direction for our future work.
>
>
> ---
>
> We hope these responses sufficiently address the reviewer’s concerns and are happy to provide further clarification if needed. We sincerely thank the reviewers once again for the thoughtful feedback, which has helped us further improve this work.
>
> ---
>
> Rererences:
>
> [1] Pippi V, Quattrini F, Cascianelli S, et al. Zero-Shot Styled Text Image Generation, but Make It Autoregressive. In CVPR 2025.
>
> [2] Chen Y, Niu Z, Ma Z, et al. F5-tts: A fairytaler that fakes fluent and faithful speech with flow matching. In ACL 2025.
>
> [3] Tan Z, Liu S, Yang X, et al. Ominicontrol: Minimal and universal control for diffusion transformer. In ICCV 2025.
>
> [4] Chen B, Zhao M, Sun H, et al. XVerse: Consistent Multi-Subject Control of Identity and Semantic Attributes via DiT Modulation. arXiv preprint arXiv:2506.21416, 2025.

---

> ### Comment · Reviewer_Kr6c · 2025-11-24
>
> I thank the authors for addressing my questions, particularly the speedup question, and the additional experiments regarding the larger vocabulary / image-based conditioning. I am happy maintaining my current rating.

---

> > ### Author Response · Authors · 2025-11-25
> > **Response to Reviewer Kr6c**
> >
> > We sincerely appreciate your positive evaluation of our work. The comments and feedback you provided are highly valuable and will help us further improve the quality of the paper.
> >
> > Best regards.

---

### Official Review · Reviewer_p6M8 · 2025-10-31

**Soundness:** 3
**Presentation:** 4
**Contribution:** 3
**Rating:** 6
**Confidence:** 2

**Summary:**

This paper addresses the challenge of text-to-online handwriting generation (TOHG), focusing on full-line handwritten Chinese text. It introduces a latent diffusion Transformer framework for full-line handwriting synthesis, featuring InkVAE for latent-space modeling and InkDiT for precise pen trajectory generation. Experiments show that DiffInk surpasses current SoTA methods in both glyph accuracy and style fidelity for TOHG.

**Strengths:**

1. This paper proposes the glyph- & style-aware regularization to preserve character-level structural information and enhance the discriminability of writing styles. The introduced InkVAE can learn more discriminative and representative latent features.
2. The proposed InkDiT performs the diffusion process on the latent space, which is more efficient and effective than previous Diffusion models for TOHG.
3. Experiments show that the proposed DiffInk outperforms the previous SoTA models for TOHG with higher glyph accuracy, style fidelity, and generation efficiency.

**Weaknesses:**

1. The competing models in Table 1 are slightly few, with only three different models. More competing models for OHG should be included, such as SketchRNN.
2. Although TOHG is less investigated, the generation paradigm of DiffInk is very similar to the existing latent generation model StableDiffusion.

**Questions:**

1. InkDiT mainly performs diffusion on latent representations, and thus DiffInK heavily relies on the performance of VAEDecoder in InkVAE. The question is whether InkVAE can generalize to unseen writers and text corpora, especially for full-line text generation?

2. Although InkVAE can learn glyph- & style-aware features, the style and glyph information of each handwriting sample are mixed or entangled in latent presentations (extracted by InkVAE). In other words, the style features and textual contents are entangled, and only paired styles & texts can successfully reconstruct the input handwriting sample. Therefore, the concern is whether the X_ref and Content Z in Fig. 3 must attain the overlapped textual contents? Whether the model capable of generating accurate handwriting samples if the textual contents of X_ref and Content Z are completely disjointed?

3. Different from the auto-regressive model that explicitly produces the end-of-sequence, how does the DIffInk know the lengths of generated full-line handwriting trajectories?

---

> ### Author Response · Authors · 2025-11-24
> **Response to Reviewer p6M8 [1/2]**
>
> We thank the reviewer for the thoughtful comments and valuable feedback, particularly the positive assessment of our overall approach, the latent-space modeling capability of InkVAE, and the advantages of DiffInk in glyph accuracy, style fidelity, and generation efficiency.
>
> We address the reviewer’s concerns and questions as follows.
>
> ---
> - *Q1: Can InkVAE generalize to unseen writers and unseen text content?*
>
> **Yes. All experiments are conducted on the test set, which contains writer styles and text-line contents that are entirely non-overlapping with those in the training data.**
>
> InkVAE demonstrates strong generalization ability, as its inference process does not rely on writer identity or specific text content. The writer-classification loss and character-recognition loss introduced during training serve purely as latent-space regularization, encouraging the model to learn disentangled and discriminative representations of style and content. Furthermore, InkVAE not only generalizes well to unseen writing styles and text-line content, but also enhances the performance of downstream tasks. As shown in the table, InkDiT achieves state-of-the-art performance when conditioned on the latent space provided by InkVAE, whereas removing the OCR loss or the style-regularization loss leads to noticeable drops in both character accuracy and style scores. This demonstrates that a semantically enriched latent space facilitates more effective conditioning for the diffusion model, thereby further validating the design of InkVAE.
>
> ### Ablation Study of InkVAE and Its Impact on InkDiT Generation
>
> | **VAE Losses** |  |  | **VAE Reconstruction Performance** |  |  |  | **InkDiT Generation Performance** |  |  |  |
> |----------------|--|--|------------------------------------|-----------|-----------|-----------|------------------------------------|-----------|-----------|-----------|
> | **L_rec+kl** | **L_ocr** | **L_sty** | **AR% ↑** | **CR% ↑** | **Style ↑** | **DTW ↓** | **AR% ↑** | **CR% ↑** | **Style ↑** | **DTW ↓** |
> | ✓ | × | × | 97.59 | 97.61 | 99.97 | 0.014 | 74.77 | 75.20 | 60.68 | 1.062 |
> | ✓ | ✓ | × | 97.60 | 97.63 | 99.97 | 0.015 | 82.09 | 82.41 | 66.07 | 1.052 |
> | ✓ | × | ✓ | 97.59 | 97.62 | 99.98 | 0.016 | 79.64 | 79.95 | 68.99 | 1.059 |
> | ✓ | ✓ | ✓ | 97.65 | 97.67 | 99.97 | 0.016 | **94.38** | **94.58** | **77.38** | **1.049** |
>
> ---
> - *Q2: Regarding the concern about the content input*
>
> We sincerely thank the reviewer for the careful observation and insightful feedback. **DiffInk can still generate correct results even when the text content Z and the reference style $X_{\text{ref}}$ are completely non-overlapping.**
>
> **Generation with Randomized $T_{\text{ref}}$:** To address the reviewer’s concern, we conducted additional experiments in which the initial character labels $T_{\text{ref}}$ were randomized. The quantitative results are presented in the table mark as **DiffInk-Random**. As shown, both content and style scores exhibit a slight decrease but still remain higher than those of the baseline methods.
>
> **Generation with OCR-Derived $T_{\text{ref}}$:** In practice, the text content of the reference handwriting $T_{\text{ref}}$ can be easily obtained using existing recognition models, so this issue has little impact on real-world applications. We also conducted experiments under this setting, and the corresponding results are reported in the **DiffInk-OCR** row of the table below, where the generated text lines achieve a content accuracy of nearly 94% and a style score of 76.
>
> **Generation with Paired $X_{\text{ref}}$ and $T_{\text{ref}}$:** During training and inference, we keep $T_{\text{ref}}$ paired with the corresponding handwriting sequence $X_{\text{ref}}$, which is not a special design choice but simply follows the standard setting adopted in prior works [1][2]. From the perspective of training dynamics, the loss function is applied only to the generation region $x_{\text{gen}}$, while no supervision is imposed on the reference portion. Consequently, the specific characters in $T_{\text{ref}}$ do not affect the generated results.
>
> ### Quantitative Generation Results under Different $T_\text{ref}$ Settings
>
> | **Methods**        | **AR% ↑** | **CR% ↑** | **Style ↑** | **DTW ↓** |
> |--------------------|-----------|-----------|-------------|-----------|
> | **Drawing (New added)**            | 76.35     | 76.87     | 25.75       | 1.582     |
> | **DeepImitator (New added)**       | 78.21     | 78.98     | 27.62       | 1.561     |
> | WLU                | 79.85     | 80.23     | 35.71       | 1.540     |
> | SDT                | 82.53     | 83.00     | 50.51       | 1.270     |
> | OLHWG              | 91.48     | 91.71     | 44.74       | 1.326     |
> | **DiffInk-Random** | 92.23     | 92.95     | 73.47       | 1.105     |
> | **DiffInk-OCR**    | 93.81     | 93.98     | 76.29       | 1.067     |
> | **DiffInk (Ours)** | **94.38** | **94.58** | **77.38**   | **1.049** |

---

> ### Author Response · Authors · 2025-11-24
> **Response to Reviewer p6M8 [2/2]**
>
> - *Q3: Regarding the question on how the model knows when to stop*
>
> This is a valid and important question. **The generation process can determine when to stop based on the decoding results of InkVAE**. During InkVAE training, the reconstruction loss for the x and y coordinates is computed only over the valid (non-padded) region, whereas the pen-state loss is computed over the full sequence, including padding positions. As a result, the decoder of InkVAE also learns to output pen-state signals for padded regions. **The end-of-sequence can therefore be determined by detecting consecutive padding pen-state outputs**, allowing the model to identify the termination point of the generated trajectory.
>
> ---
>
> - *W1: Concern about the limited baselines.*
>
> We appreciate the reviewer’s constructive comments. In the revised version, we have **added two additional baselines**—Drawing [3] and Deep-Imitator [4]—both derived from the SketchRNN[5] and widely used for handwritten character generation. The detailed experimental results are shown in the table below, and the superiority of our method remains clearly demonstrated. We believe these experiments have made our paper more convincing, and we would appreciate any additional comparison methods the reviewers may suggest.
>
> ### Quantitative comparison with SOTA handwriting generation methods.
>
> | **Method** | **Venue** | **Output** | **AR% ↑** | **CR% ↑** | **Style ↑** | **DTW ↓** | **Avg. char/s ↑** |
> |------------|-----------|------------|-----------|-----------|-------------|-----------|-------------------|
> | **Drawing (New added)**       | TPAMI'17 | Char-level | 76.35 | 76.87 | 25.75 | 1.582 | 4.16 |
> | **DeepImitator (New added)**  | PR'20 | Char-level | 78.21 | 78.98 | 27.62 | 1.561 | 3.96 |
> | WLU           | CGF'21 | Char-level | 79.85 | 80.23 | 35.71 | 1.540 | 25.19 |
> | SDT           | CVPR'23 | Char-level | 82.53 | 83.00 | 50.51 | 1.270 | 3.35 |
> | OLHWG         | ICLR'25 | Char+layout | 91.48 | 91.71 | 44.74 | 1.326 | 0.07 |
> | **DiffInk (Ours)** | **This work** | **Line-level** | **94.38** | **94.58** | **77.38** | **1.049** | **58.47** |
>
> ---
>
> - *W2: The relation to Stable Diffusion–like paradigms*
>
> **Summary of Methodology and Contributions:**
> As discussed in the related work section, recent advances in image and speech generation highlight the central role of latent diffusion models in achieving high-quality and expressive generative results. Prior to our work, however, research on handwriting generation remained largely focused on single-character synthesis, while text-line generation was widely considered challenging due to its structural complexity and long-range dependencies. Within this research background and the challenges posed by the task, our innovations are twofold:
>
> **1) Research Motivation:**
> Beyond traditional character-level generation, we introduce a **fully end-to-end framework for text-line modeling**. DiffInk is the **first** unified approach designed for efficient, high-quality online handwritten text-line synthesis, providing a direct and conceptually novel solution to synthesizing long and complex handwriting sequences. It further advances the field of handwriting synthesis from isolated character generation to full text-line generation, offering new insights and directions for future research in this community.
>
>
> **2) Technical contributions:**
> **Designing an efficient VAE for downstream conditional synthesis of handwritten text lines is a non-trivial task**, as demonstrated in the VAE ablation analysis presented in **Response [1/2]**, directly combining a vanilla VAE with a conditional diffusion model yields only moderate character accuracy and style consistency.To address this issue, we **introduce novel glyph- and style-based regularization strategies within InkVAE**, which strengthen the latent structure and improve the stability of downstream latent diffusion generation. Building upon InkVAE, we propose InkDiT, a conditional latent diffusion Transformer that synthesizes stylistically faithful, high-quality handwritten text lines based on textual inputs and reference styles. Extensive comparisons and ablation studies validate the soundness and effectiveness of our design. As noted by other reviewers, the task-related regularization mechanisms introduced in InkVAE offer broadly useful insights that can benefit other handwriting-generation tasks.
>
> ---
>
> We hope these responses sufficiently address the reviewer’s concerns and are happy to provide further clarification if needed. We sincerely thank the reviewers once again for the thoughtful feedback, which has helped us further improve this work.

---

> ### Author Response · Authors · 2025-11-24
> **Response to Reviewer p6M8**
>
> References
>
> [1] Pippi V, Quattrini F, Cascianelli S, et al. Zero-Shot Styled Text Image Generation, but Make It Autoregressive. In CVPR 2025.
>
> [2] Chen Y, Niu Z, Ma Z, et al. F5-tts: A fairytaler that fakes fluent and faithful speech with flow matching. In ACL 2025.
>
> [3] Zhang X Y, Yin F, Zhang Y M, et al. Drawing and recognizing chinese characters with recurrent neural network. In TPAMI 2017.
>
> [4] Zhao B, Tao J, Yang M, et al. Deep imitator: Handwriting calligraphy imitation via deep attention networks. In Pattern Recognition 2020.
>
> [5] Ha D, Eck D. A neural representation of sketch drawings. arXiv preprint arXiv:1704.03477, 2017.

---

### Author Response · Authors · 2025-12-03
**General Response [1/3]**

We sincerely thank all reviewers for their time and for providing insightful and professional feedback, and especially for their active participation in the discussion, which has significantly refined and enhanced the quality of this manuscript. We also appreciate the Area Chair for coordinating the discussion process.

Below, we summarize this work from two perspectives: (1) the motivation, technical contributions, and experimental strengths of the paper, and (2) the overall rebuttal progress and how major reviewer concerns were addressed.
The additions based on the reviewers’ comments are **highlighted in blue** in the revised version.

---

## **Overview of the Work**

This paper primarily focuses on the Text-to-Online Handwriting Generation (TOHG) task, which aims to synthesize realistic pen trajectories conditioned on textual input and style references.

### **1. Research Motivations**

- **Research Status:** Prior work **primarily focus on character-level** generation. In contrast, **full text-line generation is both important and technically challenging**, as consistently emphasized by all reviewers.

- **Contributions:** We propose DiffInk, the **first end-to-end framework** for TOHG.
    Reviewers (Kr6c, U8jE, 7DmP) acknowledged this as a significant advancement for the community.

- **Open Source:** The code will be **publicly available** to enable subsequent research in the community, which was highly encouraged by Reviewer Kr6c.

---

### **2. Key Technical Innovation**

- **DiffInk Framework:** We introduce DiffInk, the **first end-to-end text-line generation method** based on a latent space diffusion model. DiffInk uses InkVAE to efficiently model long sequence data, then employs InkDiT for the conditional handwriting text-line generation process.

- **Glyph- \& Style-Aware InkVAE:** InkVAE enhances a standard VAE via **a task-relevant regularization strategy in the latent space.** Reviewers (p6M8, Kr6c, U8jE) acknowledged **its novelty and effectiveness** for conditional diffusion modeling, and Reviewer Kr6c stressed **its potential to generalize to related tasks.**

- **Handwriting Generation with InkDiT:** Based on the InkVAE latent space, the conditional diffusion model InkDiT models the stylistic handwriting text-line generation task. It efficiently utilizes both content features (from a lightweight encoder) and style conditions (from the InkVAE encoder), **all within this unified feature space.**

**The design motivation and technical justification** for every module in DiffInk were recognized and validated by all reviewers.

---

### **3. Strong Empirical Results and Broad Applicability**

- **SOTA performance** in content accuracy, style fidelity, and structural consistency (acknowledged by all reviewers).

- **Fast generation speed** (highlighted by Reviewers p6M8, Kr6c, U8jE).

- **Extensive ablation study** demonstrates the necessity and effectiveness of every component (acknowledged by Reviewers Kr6c, U8jE, 7DmP).

- **Broad potential downstream applicability**: OCR data augmentation (acknowledged by reviewer U8jE), personalized handwriting generation (Figure 17 in the appendix) and human–computer interaction.

---

## **Summary of Rebuttal Progress and Resolution of Reviewer Concerns**

During the rebuttal phase, we provided detailed and evidence-supported responses to all reviewers’ concerns. Although the discussion period was interrupted due to new policy constraints, the available records show that all major concerns were effectively resolved, and reviewers’ attitudes became consistently positive. Notably, one reviewer raised the score from 4 to 6, and another confirmed that all issues were resolved and maintained a strong 8. Below, we summarize the overall progression and the resolution status of each key concern.

---

### **Overall Progress of the Rebuttal (Completed before November 25)**

- **Reviewer Kr6c (8 $\rightarrow$ 8)**: Stated that all issues had been resolved and was pleased to maintain a strong score of 8.

- **Reviewer U8jE (4 $\rightarrow$ 6)**: Confirmed that the rebuttal addressed most concerns and raised the overall score from 4 to 6, increasing both contribution and soundness.

- **Reviewer p6M8 (6)**: Received detailed responses supported by additional experiments; no remaining unresolved issues were indicated.

- **Reviewer 7DmP (4)**: Acknowledged that **our rebuttal was convincing and recognized the contribution of our work to the handwriting generation field**. The reviewer also provided additional suggestions in the second round, to which we responded with further explanations and supplementary experiments. We believe our responses and experiments sufficiently address all of the reviewers’ concerns.

---

> ### Author Response · Authors · 2025-12-03
> **General Response [2/3]**
>
> ### **Resolution of Major Concerns**
>
> **1. Novelty and technical contribution (Reviewer U8jE andj 7DmP)**.
>
> - **Our Solution**: We provided detailed explanations of the motivation, novel end-to-end generation paradigm, and the key technical innovations of our method.
> - **Reviewers Feedback**: Reviewer U8jE found the clarification convincing and **raised the score from 4 to 6**. Reviewer 7DmP’s earlier misunderstandings were clarified in our second-round rebuttal, and **our technical contributions were consistently recognized by all other Reviewers p6M8, Kr6c, and U8jE**. Although some confusion remained, Reviewer 7DmP **acknowledged the value of our open-sourced work and its contribution to the community**, a point also supported by Reviewers Kr6c and U8jE.
>
>  **2. Insufficient Baseline Comparisons (Reviewers p6M8 and 7DmP)**.
>
> - **Our Solution**: We added two additional baselines (Drawing, TPAMI 2017; DeepImitator, PR 2020) and provided both quantitative and qualitative comparisons in the revised version (Table 1 and Figure 4 in the main paper).
> - **Reviewers Feedback**: Reviewer 7DmP agreed that the expanded comparisons **resolved the concern**. Although Reviewer p6M8 did not provide a follow-up response, the newly added baselines were those they recommended, and we believe they sufficiently address the reviewer’s concerns.
>
>  **3. Use of Synthetic Data (Reviewers U8jE and 7DmP)**.
>
> - **Our Solution**: We conducted additional experiments showing that synthetic data mitigates real-data limitations (e.g., long-tail distributions and limited volume), improves robustness, and enhances content accuracy and style consistency. A brief fine-tuning stage on real data further eliminates side effects.
> - **Reviewers Feedback**: Reviewers U8jE stated that their **concerns had been satisfactorily addressed.** Reviewer 7DmP acknowledged the **reasonable use of data augmentation** and expressed interest in future exploration of this aspect.
>
> **4. Generation Speed and Efficiency (Reviewer Kr6c and 7DmP)**.
>
> - **Our Solution**: We clarified the measurement protocol in the appendix and added an ablation study on diffusion steps for both OLHWG (ICLR 2025) and our method, as requested.
> - **Reviewers Feedback**: Reviewer Kr6c confirmed the clarification **resolved the concern**. Reviewer 7DmP’s additional requests would be **addressed through the new experiments**.
>
> **5. Availability of Reference Text Input (Reviewer p6M8)**.
>
> - **Our Solution**: We added experiments demonstrating that the model can still generate correct outputs without $T_{ref}$, and noted that $T_{ref}$ can be obtained automatically via OCR when needed.
> - **Reviewers Feedback**: Although Reviewer p6M8 did not provide a follow-up response, the **additional results addressed the concern with no further objections raised**.
>
> **6. Image-Based Text Conditioning (Reviewer Kr6c)**.
>
> - **Our Solution**: We performed a comparison experiment showing that the image-based approach currently underperforms, particularly in encoding quality and alignment, reinforcing the advantages of our design.
> - **Reviewers Feedback**: Reviewer Kr6c found the additional analysis **resolved the concern** and noted that it **further strengthened the paper**.
>
> **7. Fairness of Layout Prediction for Baselines (Reviewer U8jE)**.
>
> - **Our Solution**: We re-test single-character baselines using ground-truth text-line layouts before concatenation, followed by quantitative evaluation. Our method still outperformed all baselines.
> - **Reviewers Feedback**: Reviewer U8jE confirmed the concern was fully resolved and **increased the score from 4 to 6.**
>
> **8. Limited Exploration of Downstream Tasks (Reviewer U8jE)**.
>
> - **Our Solution**: We reproduced an additional OCR model (GLRNet, ICDAR 2021) and demonstrated that our generated data significantly improves OCR performance as a data-augmentation strategy.
> - **Reviewers Feedback**: Reviewer U8jE found this is satisfactory and **raised the score from 4 to 6** accordingly.
>
> **9. Concerns about InkVAE’s generalization ability and stopping mechanism (Reviewer p6M8)**.
>
> - **Our Solution**: The generalization capability of InkVAE is well supported by our experiments, where the **test set is completely disjoint from the training set** in both writer styles and textual content. Regarding the sequence stopping mechanism, we explicitly design a dedicated loss component within InkVAE to ensure stable and reliable prediction of the end-of-sequence.
> - **Reviewers Feedback**: Although Reviewer p6M8 did not provide a follow-up response, the **this response addressed the concern with no further objections raised**.

---

> ### Author Response · Authors · 2025-12-03
> **General Response [3/3]**
>
> **10. Availability of Mixing Style Inputs (Reviewer Kr6c)**.
>
> - **Our Solution**: Additional experiments (Figure 11 in the appendix) show that DiffInk can handle more complex style inputs and generate text-line outputs whose styles extend beyond those observed in the training set while preserving accurate content.
> - **Reviewers Feedback**: Reviewer Kr6c **confirmed this resolved the concern and strengthened the paper.**
>
> **11. Concerns about baseline comparisons (Reviewer 7DmP).**
> - **Our Solution**: We converted the baseline methods’ generation results from the online modality to the offline modality and evaluated them using a commercial OCR system. The results show that, even under this setting, the baseline methods still achieve lower recognition accuracy compared to ours.
> - **Reviewers Feedback**:  Although Reviewer 7DmP did not provide a follow-up response, the **additional results addressed the concern with no further objections raised**.
>
> ---
>
> We sincerely appreciate the reviewers’ constructive suggestions, which have helped strengthen the clarity and overall quality of our paper.

---

### Meta-Review · Area_Chair_eaV5 · 2026-01-01

**Summary:**

This submission proposes DiffInk, a latent diffusion Transformer framework for text-to-online handwriting generation (TOHG) at the full text-line level. The method contains two main components: InkVAE, a sequential VAE enhanced with OCR-based and style-classification latent regularization losses, and InkDiT, a conditional latent diffusion Transformer that generates coherent pen trajectories conditioned on target text and a reference style. Reviewers generally agree that the paper is clearly written, the problem setting is meaningful and underexplored, and the empirical results are strong in terms of glyph accuracy, style fidelity, and generation efficiency.

Across the discussion, multiple key concerns were raised around novelty/positioning, baseline coverage, synthetic data effects, evaluation fairness, and efficiency reporting. The rebuttal provides concrete additional experiments and clarifications addressing many of these issues, and at least one reviewer explicitly increased their score after the rebuttal. While one reviewer remained unconvinced about the degree of algorithmic novelty, the work is solid on the evaluated setting and the rebuttal substantially improved clarity and evidence on protocol/fairness questions. Overall, the paper presents a well-motivated and empirically supported approach on the stated experimental scope for end-to-end text-line online handwriting synthesis.

**Reviewer Concerns:**

### Concerns addressed (partially or fully) by the rebuttal

- Baseline coverage: A reviewer noted that Table 1 included too few competing methods (e.g., SketchRNN-like). The authors added Drawing (TPAMI 2017) and DeepImitator (Pattern Recognition 2020) and provided updated quantitative comparisons.

- Novelty / relation to latent diffusion paradigms: In response to concerns that the approach may be incremental or resemble existing latent diffusion pipelines, the authors clarified that the paper targets end-to-end full-line online trajectory generation, and highlighted two task-specific components: InkVAE’s latent regularization (OCR + style losses) and InkDiT’s conditional diffusion design. They supported this with ablations showing that using a vanilla VAE latent space + diffusion yields substantially weaker generation quality.

- Synthetic data and potential side effects: Reviewers questioned whether stitched-character line synthesis could harm continuity and distort the learning objective. The rebuttal added experiments comparing training with vs. without augmentation, reporting large improvements in AR/CR and style; it also reported that a brief real-data fine-tuning stage can further improve style/DTW.

- Evaluation fairness for character-level baselines: To address concerns that using a shared layout predictor may disadvantage character-level baselines, the authors re-tested these baselines using ground-truth bounding boxes for stitching and reported updated results, while noting DiffInk remains stronger.

- Efficiency measurement protocol and DDIM-step tradeoff: The authors clarified the speed protocol (batch size 1, same RTX A6000, includes sampling and VAE decoding, excludes data loading/preprocessing) and added DDIM step ablations. They also reported an experiment accelerating OLHWG by switching from DDPM to DDIM and compared speed/quality under matched DDIM steps.

- Style mixing / more complex style conditioning: In response to questions about complex style inputs, the authors added a feature-level style mixing experiment via weighted averaging of latent style representations.

- OOV / alternative text conditioning: For concerns about a fixed character codebook and OOV handling, the authors conducted a preliminary study on image-based glyph conditioning (ResNet-18) under two schemes and reported results, which underperformed the embedding-based conditioning in their setting.

- Reference text availability / disjoint content between style reference and target content: The rebuttal included experiments using randomized reference labels and OCR-derived reference labels, showing the method remains strong relative to baselines even when reference content does not overlap with the target.

### Concerns still outstanding (not decision-critical)

- Breadth of evaluation across datasets/domains: One reviewer noted evaluation is primarily on a single Chinese dataset and suggested broader testing (e.g., other languages, sketches, math). This remains a limitation of the current submission.

- Strength of downstream OCR evaluation: A reviewer requested validation on stronger OCR systems. The authors responded by retraining their OCR baseline on the full dataset and reporting improved results with DiffInk-generated augmentation, but fully establishing gains on state-of-the-art text-line OCR systems remains constrained by limited open-source availability.

**Reviewer Scores:**

- Kr6c: 8 (maintained; explicitly stated they are happy maintaining rating after rebuttal clarifications and added experiments)

- p6M8: 6 (marginally above threshold)

- U8jE: 4 → 6 (explicitly raised score after rebuttal and additional experiments; later provided further suggestions)

- 7DmP: 4 (maintained; acknowledged parts resolved but remained unconvinced about novelty; still stated the work offers value to the community)

---

### Decision · Program_Chairs · 2026-01-26

Accept (Poster)